# Will daytime community calcification reflect reef accretion on future, degraded coral reefs?

Coulson A. Lantz[1,2], William Leggat[2], Jessica L. Bergman[1], Alexander Fordyce[2], Charlotte Page[1], Thomas Mesaglio[1], Tracy D. Ainsworth[1]

[1]University of New South Wales, School of Biological, Earth and Environmental Sciences, Kensington, 2033 NSW Australia

[2]University of Newcastle, School of Environmental and Life Sciences, Callaghan 2309 NSW Australia

Email Corresponding Author: C.lantz@unsw.edu.au

**Abstract**

Coral bleaching events continue to drive the degradation of coral reefs worldwide, causing a shift in the benthic community from coral to algae dominated ecosystems. Critically, this shift may decrease the capacity of degraded coral reef communities to maintain net positive accretion during warming-driven stress events (e.g., reef-wide coral bleaching). Here we measured rates of net ecosystem calcification (NEC) and net ecosystem production (NEP) on a degraded coral reef lagoon community (coral cover < 10 % and algae cover > 20 %) during a reef-wide bleaching event in February of 2020 at Heron Island on the Great Barrier Reef. We found that during this bleaching event, rates of NEP and NEC across replicate transects remained positive and did not change in response to bleaching. Repeated benthic surveys over a period of 20 d indicated an increase in the percent area of bleached coral tissue, corroborated by relatively low Symbiodiniaceae densities ($\sim 0.6 \times 10^6$ cm$^{-2}$) and dark-adapted photosynthetic yields in photosystem II of corals ($\sim 0.5$) sampled along each transect over this period. Given that a clear decline in coral health was not reflected in the overall NEC estimates, it is possible that elevated temperatures in the water column that compromise coral health enhanced the thermodynamic favourability for calcification in other ahermatypic benthic calcifiers. These data suggest that positive NEC on degraded reefs may not equate to the net positive accretion of complex, three-dimensional reef structure in a future, warmer ocean. Critically, our study highlights that if coral cover continues to decline as predicted, NEC may no longer be an appropriate proxy for reef growth as the proportion of the NEC signal owed to ahermatypic calcification increases and coral dominance on the reef decreases.

## 1. Introduction

Coral have long been the focus of climate change research in tropical oceans, as they are a keystone species responsible for the biogenic construction of complex reef habitat (Grigg and Dollar, 1990). Adverse effects to their ability to construct calcium carbonate structure have negative implications for coral reef ecosystems, given corals are the major organism responsible for collectively maintaining the accumulation of permanent reef structure at a rate that overcomes the biological and physical mechanisms that act to break reefs down (carbonate dissolution, bioerosion, storm activity; Eyre et al., 2018). In contrast to coral-derived calcium carbonate, other benthic marine calcifiers, such as non-sessile Gastropods, Echinoderms, or Halimeda algae (Ries et al., 2009; Harney and Fletcher, 2007), secrete calcium carbonate that is relatively temporary and does not contribute to the long-term reef structure. Traditionally, corals are classed as the dominant calcifier on tropical coral reefs, occupying between 10 – 50 % of benthic area in healthy coral reef lagoons (Bruno and Selig, 2007; Brown et al., 2018). As such, estimates of net ecosystem calcification (NEC) are considered synonymous with the growth and function of the entire coral reef community and can be used to represent the collective response in coral reef community health to anthropogenic stressors such as ocean warming and subsequent reef-wide bleaching events (Courtney et al., 2018).

Presently, records of coral reef NEC during a reef-wide bleaching event (driven by sea surface temperatures + 1 °C above monthly maximum means; Heron et al., 2016; Sully et al., 2019) are rare (McMahon et al., 2019). The effects of bleaching events, and their associated thermal seawater temperature anomalies, on coral reef NEC have been predominately studied *ex-situ* using recreated communities in aquaria (Dove et al., 2013) or scaling up the response from organism-level studies, both *ex-* (Castillo et al., 2014) and *in-situ* (Cantin et al., 2010). In studies conducted *ex-situ* in aquaria, a warming treatment strong enough to cause bleaching (between 1 – 4 °C above the summer mean) reduced coral calcification rates by 30 to 90 % (Cantin et al., 2010; D'Olivo and McCulloch, 2017).

*In-situ* observations following bleaching events have shown a 20 – 90 % reduction in individual coral calcification rates (Castillo et al., 2014) and a significant reduction in the coral endosymbiont photosynthetic yields (evidence of damage to their photosystems; Warner et al., 1999). At the whole community level, the few *in-situ* studies that have observed community metabolism during a bleaching event recorded a 40 % (DeCarlo et al., 2017; Dongsha Atoll, Taiwan) to 100 % (Courtney et al., 2018; Kaneohe Bay, Hawai`i; Kayanne et al., 2015; Palau) decline in reef NEC. This effect has been observed to linger six to twelve months after these events, with NEC remaining depressed by as much as 40 – 46 % (Lizard Island; McMahon et al., 2019) and an ultimate loss of 30 – 90 % of the benthic coral cover (Brown and Suharsono, 1990; Baird et al., 2002).  Experiments with simulated communities in aquaria (e.g., Dove et al., 2013) validate these organism- and community-level *in-situ* studies, where this same magnitude of warming lead to a reduction in the experimental community coral cover by 30 %, a 70 % decline in NEC, and subsequent out-competition of corals by neighbouring algae.

The overgrowth of algae has been mirrored in the natural reef lagoon environment several times following bleaching events (Hughes et al., 1999; Diaz-Pulido et al., 2009). Despite a recovery to normal pre-disturbance NEC within two years following a 2014 bleaching event at Lizard Island (Pisapia et al., 2019), there was a permanent shift from coral to algae as the dominant benthic community member, with a decline in coral cover from 8 % to 3 % along transects established at the southeast end of the lagoon (McMahon et al., 2019). This response has been seen elsewhere on the Great Barrier Reef, where reef-wide bleaching events lead to the overgrowth of unpalatable *Lobophora vareigata* algae (Diaz-Pulido et al., 2009) to the extent that coral became a minority constituent (~ 2 – 5 %) in the lagoon's benthic community. This transition to an algal-dominated reef community jeopardizes the efficacy of NEC as a proxy for reef growth given that hermatypic corals can no longer be considered the dominant benthic organism (Courtney et al., 2018). Similar questions have been raised after other anthropogenically-driven stress events (e.g., eutrophication and sedimentation; Edinger et al., 2000) where coral growth rates on undisturbed reefs did not differ from those measured

on polluted, algal-dominated reefs where habitat structure was clearly degrading. If the community predominantly becomes covered in algae and the habitat structure is visibly degrading, does NEC still represent reef growth or does it now reflect a greater proportion of ahermatypic organism calcification not contributing to permanent structure?

Shift from coral to algal dominated reefs without the concomitant decline in NEC have been observed by Kayanne et al., (2005; 7.1 % coral cover), where no change in NEC on Shiraho Reef, Japan was measured despite 51 % of the corals bleaching during a 1998 bleaching event and a decline to 5.8% coral cover. This study suggested that continued calcification by living, unbleached corals, calcifying algae, or other benthic calcifiers (e.g., foraminifera, gastropods, echinoderms) may have compensated for any expected bleaching-driven decline in coral calcification. This discrepancy between Kayanne et al., (2005; no change in NEC on a reef with < 10 % coral cover) and that of other NEC estimates during a bleaching event (decline in NEC on a reef $\geq$ 10 % coral cover; DeCarlo et al., 2014) may be due to a critical threshold in the relationship between NEC and percent coral cover. This is of specific concern when using NEC to monitor community function (i.e., the net accretion of reef structure) during coral bleaching or other disturbance events on future, degraded reefs where algae will likely become the dominant benthic member.

To address these emerging concerns, this study investigated community metabolism on a degraded coral reef community (coral cover < 10 %, algae cover > 20 %) during a bleaching event at Heron Island on the Great Barrier Reef in February of 2020. Flow-metabolism transects were established on two areas within the Heron Island lagoon and estimates of community metabolism (NEP and NEC), coral metaorganism function (photosynthetic yields, Symbiodiniaceae densities), benthic cover, and bleaching extent (percent bleached coral tissue) were assessed during the period of peak thermal stress.

**2. Materials and Methods**

**2.1 Study Area**

This study was conducted from January 15th to February 10$^{th}$ of 2020. Two separate 200m x 100m
lagoon sites (Lagoon site 1 and 2; Fig.1) that each differed in total coral cover were established on the
southern side of the Heron Island lagoon (23° 26'670' S, 151° 54.901' E). Community metabolism,
physiochemical data, benthic community cover, and bleaching extent were then repeatedly measured
on each transect over a period of 20 days. HOBO temperature loggers (Onset, USA), which recorded
temperature (ºC) at an interval of 15 minutes, were deployed at nine upstream and downstream
locations (1 - 9) across the study area (Fig. 1). Overlapping loggers located at the middle deployment
locations (2, 5, and 8) were used for both Lagoon site 1 and 2, resulting in six loggers per site.
To measure the accumulation of temperature stress above the local bleaching threshold (defined here
as the Maximum of the Monthly Means, MMM + 1 = 28.3 °C; Liu et al. 2014;) mean temperatures
across all nine loggers were used to calculate the number of Degree Heating Weeks (DHWs), which
represents the 12-week accumulation of temperatures above the MMM (Heron et al., 2016). Because
HOBO temperature loggers may record higher temperatures than surrounding seawater due to internal
heating of the transparent plastic casing (Bahr et al., 2016), HOBO loggers were deployed in the shade
on a cinderblock and downloaded temperature data were corrected for precision (48-h side-by-side
logging of all nine loggers in an aquarium) and accuracy (deployment next to Hanna HI98194
multimeter recording temperature). Light loggers ($2\pi$ Odyssey PAR sensor) were deployed within the
middle of each study site (n = 1 site$^{-1}$). Loggers were attached to a star picket to ensure the sensor was
exactly 20 cm above the benthos and recorded light intensity at 15-minute intervals. Odyssey light
logger data were converted to μmol quanta of photosynthetic active radiation (PAR) m$^{-2}$ s$^{-1}$ using a
linear calibration over a 24-h period with a $2\pi$ quantum sensor LI-190R and a LiCor LI-1400 meter
($R^2$ = 0.92).
**2.2 Benthic Community Surveys**
The benthic community along each 200 m transect was described using four survey approaches: 1)
Point-contact surveys, 2) Photo-quadrat surveys, 3) Mobile invertebrate counts, and 4) Invertebrate
and algal taxonomy descriptions. For the 1) Point-contact surveys and 2) Photo-quadrat surveys,
benthic cover was categorized as coral (hermatypic, live), coral (bleached), coral (soft), algae (fleshy,
non-calcifying), other calcifier (e.g., *Halimeda* spp.), rubble, and sediment. For the point-contact
method, the occupier of benthic space was recorded underneath each 1 m interval (n = 200 transect$^{-1}$)
at the beginning and end of the study and data are presented as relative % cover.  These surveys were
repeated twice per transect at the beginning of the study (Jan 18-20 2020) to provide an initial
understanding of the community assemblage prior to flow-metabolism measurements.  For the 2)
photo-quadrat method, a photo of a 1 m$^2$ PVC quadrat was taken at every 5 m interval (n = 40 transect$^{-1}$
) three times throughout the study: 1) at the beginning prior to any observed bleaching (Jan 24 2020),
2) in the middle after the first observed bleaching event (Feb 6 2020), and 3) at the end of the study
after several more observed bleaching incidents (Feb 13 2020).. These images were analysed in ImageJ
using one side of the photo quadrat to set the scale (1 m) and the area tracing tool calculate the relative
% area of each category over time.
For mobile invertebrate surveys, a transect tape was laid along each 200 m transect length relatively
large, easily visible mobile invertebrates (e.g., sea cucumbers, sea hares, sea urchins) located 1 meter
to the left or right along the transect were counted. Surveys were conducted at dawn to ensure a balance
of visibility and invertebrate activity and repeated 3 times along each transect (n = 9 site$^{-1}$). Data are
presented as abundance counts per m$^2$ (individuals m$^{-2}$). Individuals present at less than 0.1 m$^{-2}$ were
excluded from the final data reported but were included as part of the invertebrate taxonomy described
below. For general invertebrate taxonomy, while conducting the survey approaches detailed above,
each time a new invertebrate morphospecies was encountered, photographs were taken and uploaded
to iNaturalist, a biodiversity citizen science platform where identifications are contributed in real time
by both amateur naturalists and professional taxonomists as part of a consensus system
([www.inaturalist.org](http://www.inaturalist.org)). Using a combination of taxonomic keys and crowdsourcing via iNaturalist,
algae, corals, and other sampled marine invertebrates were identified to as fine a taxonomic level as
possible. These data are presented as presence/absence across the entire 200 m x 400 m study area.
Because sampling was conducted at low tide, most fish usually present in the lagoon were absent and
excluded from benthic survey data.

**2.3 Bleached Coral Physiology**

Following the qualitative appearance of bleaching (white corals in photo quadrat surveys), efforts were
made to provide physiological data that would corroborate bleaching observations. This was
accomplished through Symbiodiniaceae density analyses for both *Acropora* spp. (*Acropora aspera,*
*Acropora millepora, Acropora muricata, Acropora humilis*) and "Other" corals (*Pocillopora*
*damicornis, Isopora palifera, Porites cylindrica, Montipora digitata*). For photophysiology, replicate
coral fragments (n = ~15 - 35 time point$^{-1}$) of both Acropora spp. and "Other" corals were collected
across all transects at Lagoon site 1 and 2 by hand on Feb 4 and Feb 9, 2020 (once bleaching was
apparent) and used to measure photosynthetic efficiency of in hospite Symbiodiniaceae cells.
Measurements of photosystem II dark-adapted yield were taken using a Pulse-Amplitude Modulated
(PAM) fluorometer (MAXI Imaging PAM, Waltz, Effeltrich, Germany) using imaging PAM analysis
(n = 3 technical replicates per fragment).
For quantification of Symbiodiniaceae densities, replicate coral fragments (n = ~15 - 35 time point$^{-1}$)
of both Acropora spp. and "Other" corals  were collected across all transects at Lagoon site 1 and 2 by
hand on Jan 30 and Feb 12 2020. At each sampling time points the most visually 'stressed' (ranging
from pale to completely bleached) corals were collected. 15 fragments from each group (Acropora spp.
or "Other") were collected at the study site and directly frozen in WhirlPak$^{©}$ bags at -80 °C. Tissue
was removed from the skeleton using an airpik and compressed air from diving tanks. Tissue was
blown into a zip-lock bag with 50ml of 0.45 μ filtered seawater. The algal pellet was washed three
times (centrifuged at 3856 x g, 4 °C for 5 minutes) to remove mucous and coral tissue, before being
frozen at -20 °C for later analysis. The pellet was suspended in 10 ml of filtered sea water and aliquots
were counted in triplicate using an improved Neubauer haemocytometer. Counts were normalized to
fragment surface area using the wax method (Stimson and Kinzie III, 1991).

**2.4 Lagoon Community Metabolism Measurements**

Rates of daytime net ecosystem production (NEP; mmol $O_2$ $m^{-2}$ $h^{-1}$) and net ecosystem calcification
(NEC; mmol $CaCO_3$ $m^{-2}$ $h^{-1}$) were estimated daily (tides and full sunlight permitting) over the course
of 20 d (Jan 22 to Feb 12 2020) along the six transects. To estimate rates of NEP and NEC, changes in
dissolved oxygen (DO) and total alkalinity ($A_T$) were measured, respectively, during a three-hour
window around low tide and peak sunlight using both the slack-water and flow-respirometry (Eulerian)
approach. Because differences in sunlight are a major driver in NEP variability, measurements were
refined to days of full sunlight and low tides coinciding with near mid-day (11:00 – 15:00). Flow
speeds across the transect were measured with an acoustic doppler velocimeter (ADV; Sontek [cm $s^{-1}$
]) recording data at 15-min intervals. This ADV was placed at the end of the middle transect (Figure
1). Depth varied between 0.1 – 1m and was measured concurrently with water sample collections at
each location. Depth was also measured at peak low tide at 5m intervals along each transect (n = 120
site-1) to ensure that sample location depths adequately represented the entirety of the transect.
Salinity (psu) and dissolved oxygen (DO: mg $L^{-1}$) was measured with a Hanna HI98194 multimeter
and DO was converted to $\mu$mol $kg^{-1}$ using seawater density. DO probe calibration was performed
weekly using a two-point calibration at 0 % (sodium thiosulfate) and 100 % saturated seawater
equilibrated with the atmosphere. Samples for $A_T$ were collected in 60 ml sample polycarbonate
sample bottles, preserved with saturated Mercuric Chloride according to $CO_2$ best practices (Dickson,
2007), and sealed with a screw top lid and parafilm. Seawater $A_T$ was analysed by potentiometric
titration using a Metrohm 848 Titrino plus automatic titrator (~ 40 ml of seawater per sample) in
duplicates (SD uncertainty < 2 $\mu$mol $kg^{-1}$). Overall analytical uncertainty for $A_T$ (SD = $\pm$ 2.4 $\mu$mol $kg^{-1}$
) measurements was estimated from repeated measurements of certified reference materials from the
Scripps Institute of Oceanography (CRM; Batch 161).

### 2.4.1 Eulerian Approach

Flow metabolism transects were established along a reef area previously characterised as degraded, where there is less than 10 % coral cover (Roelfsema et al., 2018). The flow-respirometry (i.e., Eulerian approach) measurements were conducted within two designated reef areas (100 m x 200 m; 0.02 km$^2$) that significantly differed in coral cover. The defined study area was determined based on the necessary transect length to achieve measurable differences in seawater dissolved oxygen ($\Delta DO = \pm 4 - 7$ mg L$^{-1}$) between upstream and downstream locations (~ 200 + m; Langdon et al., 2010).

Repeated deployments of fluorescein dye packets across the research zone at differing tidal periods determined a specific 400 m x 100 m area of the reef where flow was unidirectional from east to west. This period spanned from 2 hours before to 1 hour after peak low tide (3 hours total). Outside of this period, the reef lagoon was no longer physically separated from the open ocean, flow became multidirectional, and the defined lagoon area became too deep and diluted with open ocean water to measure significant changes in seawater chemistry. The 400 m x 100 m area was then designated as two,. The spread of the dye path varied ± 25 m in a north/south direction and triplicate 200 m transects were spaced 50 m apart in parallel at each site so that NEC and NEP were averaged across the three downstream locations, representing all potential water flow paths of the overall study site area. A flow meter was rotated between downstream water sample collection locations on (n = 3 sampling location$^{-1}$) and determined continued placement of the one available ADV at the middle downstream location was adequate to represent flow speed across all three transects.. Within each area, three 200m transects were established in parallel, 50 m distance from one another (Fig. 1). Water samples were collected as close in time as possible at these fixed upstream and downstream locations (n = 3 area$^{-1}$) at peak low tide while lagoon currents were unidirectional, running east to west.

$$Equation\ 1: NEP \ = \frac{3600}{100} \times \frac{\Delta DO \times \rho \times u \times d}{l}$$

$$Equation\ 2: NEC\ = \frac{3600}{100} \times \frac{0.5 \times \Delta TA \times \rho \times u \times d}{l}$$
The Eulerian approach requires the following measurements: The change in DO and $A_T$ ($\Delta DO$ and
$\Delta A_T$; mmol kg$^{-1}$), the mean seawater density ($\rho$; kg m$^{-3}$), the mean current speed (cm s$^{-1}$), the mean
depth over the transect ($d$; meters), and the length of the transect ($l$; meters). For specific details on the
arrangement of the equations above, including the 3600/100 parameter (to convert cm s-1 to m h-1),
please refer to Langdon et al., (2010).
**2.4.2 Slack Water Approach**
The slack-water approach was used to estimate rates of NEP and NEC over a relatively larger area of
reef ($\sim 0.3$ km$^2$) during a period of three hours around low tide. This period was chosen based on initial
observations of current speed and direction that aligned with previous slack-water estimates on this
specific area of the Heron lagoon (Stoltenberg et al., 2020). Starting two hours before peak low tide,
the lagoon becomes separated from the open ocean and the current begins flowing unidirectionally
toward the lagoon outlet to the west. This unidirectional flow behaviour continues until roughly 2 hours
after peak low tide, at that time the flow begins to reverse as the tide fills back in over the reef crest.
To avoid dilution with the open ocean and changing current vector directions confounding residence
time estimates, water samples were collected from the same three locations (n = 3 day$^{-1}$) two hours
before peak low tide and one hour following.
$$Equation\ 1: NEP\ = \frac{\Delta DO \times \rho \times d}{\Delta t}$$
$$Equation\ 2: NEC\ = \frac{0.5 \times \Delta A_T \times \rho \times d}{\Delta t}$$
The slack-water approach requires the following measurements: The change in DO and $A_T$ ($\Delta DO$ and
$\Delta A_T$; mmol kg$^{-1}$), the mean seawater density ($\rho$; kg m$^{-3}$), mean depth over the transect ($d$; meters), and
time between sampling ($\Delta t$; hours). Given the time between samples (~ 3 h) and mean current speeds
(~ 20 cm s$^{-1}$), these measurements represent a transect length of roughly 2.5 – 3km of reef.
**2.4.3 Approach Comparison**
Both approaches to estimate NEP and NEC provide limitations and advantages with respect to each
other (see Langdon et al., 2010). In the Eulerian approach, the exact benthic area contributing to
measured changes in seawater chemistry is known and its constituents can be quantified and related to
the calculated rates of benthic metabolism. This approach, however, measures change in alkalinity
over a relatively smaller area and time-period. Resulting fluxes in $A_T$ ($\pm$ 30 – 60 µmol kg$^{-1}$) and DO
($\pm$ 20 – 50 µmol kg$^{-1}$) are relatively small compared to the slack-water approach, thereby providing
less confidence in calculated rates of benthic metabolism.
In contrast, the slack-water approach benefits from the relatively large changes in total alkalinity ($A_T$:
$\pm$ 100 – 200 µmol kg$^{-1}$) and dissolved oxygen (DO: $\pm$ 80 – 150 µmol kg$^{-1}$), which provides more
confidence in $A_T$ anomaly calculations and represent a large area of the reef flat relative to this study's
flow-respirometry estimates. This approach, however, lacks specificity of the exact area of reef
affecting changes in chemistry and DO fluxes are more vulnerable to gas exchange anomalies. As
such, relating metabolic rates to the benthic community provides uncertainties given daily changes in
mean current speed and, subsequently, the area of benthos reflected in the $A_T$ and DO anomaly.
Overall, the combination of both approaches can work in tandem to compensate for their respective
weaknesses. However, neither approach can accommodate dilution with the open ocean and generally
need to be conducted in full sunlight or darkness so that community metabolism does not transition
between autotrophy and heterotrophy in the middle of the measurements. For this reason, community
metabolism estimates were paused from Jan 27 – Feb 2 when peak low tide occurred around dawn and
dusk and changes in DO and $A_T$ were negligible.
**2.4.4 Air-Sea Gas Exchange Corrections**
NEP estimates were corrected for the air-sea gas exchange ($F_{O2}$) of oxygen using the gas-transfer
velocity relationships outlined by Wanninkhof (1992) and Wanninkhof et al., (2009). $F_{O2}$ was
calculated with the following equation.
$$F_{O2} = k\,K0\,(fO2_{water} - fO2_{air})$$
where k is the gas transfer velocity (calculated using and averaged daily wind speed from BOM
data), K0 is the gas transfer coefficient, $fO2_{water}$ is the concentration of seawater dissolved oxygen
(mg $L^{-1}$) at the time of the downstream measurement, $fO2_{air}$ (mg $L^{-1}$) was assumed to be 100 %
saturation at the air temperature over the 3-h measurement period (~ 8.10 mg $L^{-1}$).
**2.4.5 Statistical Analyses**
All statistical analyses were performed with the SPSS statistics software (SPSS Inc. 2013 Version
26.0). To compare measured differences in benthic cover (percent coral, percent algae, percent
bleached coral tissue, sediment overgrowth) and community metabolism (NEP and NEC) between
triplicate transects, measurement days (n = 12), and Lagoon sites (Lagoon site 1, Lagoon site 2, and
Slack Water), a one-way analysis of variance (ANOVA) model was used where transect, day, or site
was a fixed effect and measured values for percent cover, NEP, and NEC were treated as the response
variable. Results for percent cover compared among triplicate transects and Lagoon sites are displayed
in Tables S1 and S2, respectively. Before community metabolism measurements were compared,
assumptions of normality and equality of variance were evaluated with a Shapiro Wilk test (Table S4).
Results for community metabolism compared among triplicate transects, measurement days, and
Lagoon sites are displayed in Tables S5, S6, and S7, respectively. A Tukey HSD post-hoc test was
used to perform pairwise comparisons for measured NEC between Lagoon site 1, Lagoon site 2, and
the slack-water approach (Table S7). To explore relationships between NEC as a function of NEP,
Model II regression techniques were used to test for significant linear relationships (cutoff value p <

292 0.1) and an ANCOVA was used to test for differences in NEC vs. NEP slope categorized by Lagoon

293 site (Lagoon site 1 and Lagoon site 2).

**294 3. Results**

**295 3.1 Lagoon Community Assemblage**

296 Across the whole study area (Lagoon site 1 and Lagoon site 2 combined), the benthic community was

297 predominately covered by sediment (59 ± 7 %) and fleshy algae (25 ± 6 %). Coral cover (5 ± 3 %) was

298 slightly higher relative to other recorded sessile calcifiers (4 ± 1 %) and carbonate rubble covered in

299 coralline algae (5 ± 2 %). Algae was the dominant benthic organism in both Lagoon site 1 (28 ± 4 %)

300 and Lagoon site 2 (22 ± 4 %) and cover was significantly higher at Lagoon site 1 (p = 0.011) (Table

301 1). Lagoon site 2 exhibited a significantly higher coral coverage (8 ± 3 %) relative to Lagoon site 1 (3

302 ± 2 %) (p = 0.001), the majority of which were *A. aspera, A. millepora,* and *M. digitata*. A description

303 of the mobile and sessile invertebrate diversity is described in Fig. 2 and the supplemental information

304 (S.4). A full list of observed invertebrates and accompanying photos can be found at

305 https://www.inaturalist.org/projects/heron-island-survey-corals-inverts-and-algae.

306 Overall, we found 25 coral species in the lagoonal reef study area, 22 of which were hard corals and

307 three soft corals (Fig. 2; Table S8). Thirteen algae morphospecies were observed, with one identified

308 as species *Valonia ventricosa* and the rest unidentified. Across all other invertebrate taxa, 19 species

309 of echinoderms, bivalves, and polychaetes, and 24 species of crustaceans and gastropods were

310 observed. Of the 43 non-coral invertebrate species, 15 were associated with colonies of *Pocillopora*

311 corals. Sea cucumbers (e.g., *Holothuria* spp., *Stichopus* spp.) were the dominant mobile invertebrate,

312 the Lollyfish sea cucumber (*Holothuria atra*) was the most common across both Lagoon sites (1.2 ±

313 0.2 individuals m$^{-2}$). Second in abundance was the Hermann's Sea Cucumber (*Stichopus hermanni*)

314 (0.4 ± 0.1 individuals m$^{-2}$). Other notable invertebrates included Linckia sea stars (*Linckia guildingia,*

315 *Linckia laevigata*) and white-speckled sea hares (*Aplysia argus*) (all found in abundances < 0.1

individuals m$^{-2}$). The largest mobile invertebrates observed were Bailer Shell snails (*Melo amphora*)
at 30 cm in length and white-spotted hermit crabs (*Dardanus megistos*) occupying Bailer shells ($<$ 0.1
individuals m$^{-2}$).
Our observations included 8 species with a conservation status of near threatened or higher, including
the small giant clam *Tridacna maxima*, Herrmann's sea cucumber (*Stichopus herrmanni*), and 6 coral
species (*Porites attenuata*, *Acropora secale*, *Isopora palifera*, *Stylophora pistillata*, *Favites halicora*,
*Favites rotundata*). Notably, our observation of the aglajid slug *Tubulophilinopsis gardineri* is one of
just 5 from Heron Island, representing the southernmost limit of its eastern coast distribution. We also
observed an undescribed nudibranch species, a yellow-brown *Gymnodoris* (Figure 5). A complete list
of all species described can be found in the Supplemental Material (Table S8).
**3.2 Lagoon Light and Temperature**
Temperature across the Lagoon site 1 exhibited a mean value of 28.6 ± 1.5 °C and varied between a
minimum of 25.8 °C and a maximum of 34.8 °C (Table 2). Light at Lagoon site 1 exhibited a mean
value of 328 ± 247 μmol quanta m$^{-2}$ s$^{-1}$ and maximum values of 1001 μmol quanta m$^{-2}$ s$^{-1}$ (Fig. 1).
Temperature across Lagoon site 2 exhibited a mean value of 28.6 ± 1.5 °C and varied between a
minimum of 25.9 °C and a maximum of 34.6 °C. Light at Lagoon site 2 exhibited a mean value of 336
± 254 μmol quanta m$^{-2}$ s$^{-1}$ and maximum values of 969 μmol quanta m$^{-2}$ s$^{-1}$.
Satellite monitoring data (5 km pixel resolution; NOAA Coral Reef Watch) indicated the accumulation
of heat stress beginning on Feb 1 2020. Lagoon temperatures peaked three days following on Feb 4$^{th}$
(Fig. 1) at which time the first signs of coral bleaching were anecdotally observed within the study
area and on other areas of the Heron lagoon. Over the course of the study period a total of 3.59 DHWs
were accumulated. In the periods before and after the accumulation of heat stress (Feb 1$^{st}$ 2020),
Lagoon site 1 mean temperatures were 28.1 ± 1.4 °C and 29.0 ± 1.5 °C, respectively, and Lagoon site
2 mean temperatures were 28.0 ± 1.3 °C and 29.1 ± 1.5 °C, respectively. Further details on recorded
light and temperature data can be found in the supplemental information (S.5).

**3.3 Lagoon Community Bleaching Extent**


Dark-adapted yield was 0.662 ± 0.010 for *Acropora* spp. fragments and 0.576 ± 0.020 for "Other"
fragments (mean ± SE, n = 35) on Feb 4th . On Feb 9th , yield declined 35 % for *Acropora* spp. to
0.430 ± 0.014 (n = 15) and 25 % for "Other" fragments to 0.434 ± 0.018 (n = 20). Symbiodiniaceae
densities were $0.976 \pm 0.135 \times 10^6$ cm$^{-2}$ for *Acropora* spp. (n = 15) and $0.507 \pm 0.160 \times 10^6$ cm$^{-2}$ for
"Other" fragments (n = 10) on Jan 30th. On Feb 12th, *Acropora* spp. densities had declined by 48 % to
$0.504 \pm 0.0849 \times 10^6$ cm$^{-2}$ (n = 15) and by 18 % for "Other" fragments to $0.414 \pm 0.094 \times 10^6$ cm$^{-2}$ (n
= 15) (Fig. 3).
Altogether, the percentage of coral tissue exhibiting bleaching increased from 0 % to 60 ± 11 % over
the course of the three photo-quadrat survey efforts (Table 3; Fig. S.1). Reef sediment was found to
exhibit increased growth of green and red microbial biofilms, which grew in cover from 2 ± 1 % to 12
± 4 %. Coral bleaching observed during the study period was confirmed by PAM fluorometry (dark
adapted yield; Fv/Fm) and Symbiodiniaceae densities (cells x $10^6$ cm$^{-2}$) measured during observed
bleaching (S.6).

**3.4 Lagoon Community Metabolism**


The mean ± SD value of NEP and NEC at Lagoon site 1 and Lagoon site 2 (pooled together across
triplicate transects and measurement days [n = 36]) is displayed in Table 4 and Fig. 3 and separated by
the pre-bleaching (Jan 22nd to Feb 1st 2020) and bleaching period (Feb 2nd to Feb 10th  2020). Mean
daytime net ecosystem production (NEP), averaged across all days and sites, was 39.4 ± 12.2 mmol
O$_2$ m$^{-2}$ h$^{-1}$. NEP did not significantly differ across triplicate transects within Lagoon site 1 (p = 0.471)
or Lagoon site 2 (p = 0.917), so these data were pooled together to represent the overall community
NEP of each site (Fig. 3). The measured NEP throughout the study period was highly variable and did
not significantly differ over time (n = 12) at either Lagoon site 1 (p = 0.181) (lowest coral cover site)
or Lagoon site 2 (p = 0.099) (highest coral cover site). NEP did not significantly differ between Lagoon
site 1 and Lagoon site 2 (p = 0.067). NEP values were not included for the slack-water approach given
the large source of error in air-sea oxygen exchange.
Mean daytime NEC, averaged across all days and sites, was $12.2 \pm 4.5$ mmol $CaCO_3$ m$^{-2}$ h$^{-1}$. Measured
rates of daytime NEC did not significantly differ across triplicate transects within Lagoon site 1 (p =
0.471), Lagoon site 2 (p = 0.917) or the slack water (p = 0.581), so these data were pooled together to
represent the overall NEC of each area (Table 4). Measured NEC was also highly variable and did not
significantly differ over time at Lagoon site 1 (p = 0.506), Lagoon site 2 (p = 0.365), and the slack
water (p = 0.073). Estimated NEC in the slack-water approach was significantly lower compared to
Eulerian estimates at Lagoon site 1 (p = 0.010) and Lagoon site 2 (p = 0.001); these two latter sites did
not significantly differ (p = 0.666). Changes in NEC were significantly related to changes in NEP at
both Lagoon site 1 ($r^2$ = 0.32; p = 0.042) and Lagoon site 2 ($r^2$ = 0.28; p = 0.046). Slope values for
daytime NEC vs. NEP for Lagoon site 1 and 2 were 0.28 and 0.24, respectively (Fig. S.2).
To determine potential effects of bleaching on night-time dissolution and respiration, night-time
estimates of NEC and NEP were conducted three times throughout the study near the dates of observed
progressed bleaching (Jan 23$^{rd}$, Feb 4, Feb 12$^{th}$). However, $A_T$ and DO changes were too small during
the Lagoon site 1 and Lagoon site 2 Eulerian estimates, so night-time NEC could only be confidently
calculated from slack-water estimates. We found mean slack-water nighttime NEC (- $3.1 \pm 1.1$ mmol
$CaCO_3$ m$^{-2}$ h$^{-1}$) did not significantly differ across transects (p = 0.617) or over time (p = 0.083) within
the current study.
**4. Discussion**
**4.1 Community Metabolism Response to Bleaching**

The southwestern lagoon area of Heron Island (southern Great Barrier Reef) is a community characterised by low coral cover of approximately 5 – 8 %. Within this reef area, the predominant benthic cover was unpalatable algae (approximately 21 %), dominated by the two genera *Laurencia* spp. and *Lobophora* spp., consistent with that of a degraded coral habitat (Hughes et al., 1999). Prior surveys of the benthic cover in this area of the Heron Island lagoon (Scientific Zone) have also estimated relatively low coral cover (0 - 10 %; Roelfsema et al., 2018).

Accumulation of heat stress in the lagoon over the study period resulted in 3.59 DHWs as *in-situ* mean temperature was elevated from ~ 28.0 °C to ~ 29.1 °C (+1.1 °C). Over this period, we found that approximately 60 % of corals present within both Lagoon sites 1 and 2 exhibited bleaching. These bleaching observations were corroborated by both photosynthetic yields and Symbiodiniaceae densities of all corals sampled. Photosynthetic yields recorded on Feb 4[th] 2020 in both the *Acropora* spp. and "other" category were barely above values considered "healthy" (0.5 [Gierz et al., 2020]) and, by Feb 9[th] 2020, exhibited symbiont loss with values below 0.5 (Acro = 0.43 ± 0.01; Other = Acro = 0.43 ± 0.01). Mean Symbiodiniaceae densities across both time points for the *Acropora* spp. (0.74 ± 0.11 x $10^6$ $cm^{-2}$) and "other" corals (0.46 ± 0.13 x $10^6$ $cm^{-2}$) were also below normally healthy values previously recorded in both *Acropora* spp. (1- 2 x $10^6$ $cm^{-2}$ [Gierz et al., 2020]) and corals in the "Other" category (e.g., *Montipora digitata*; 2-3 x $10^6$ $cm^{-2}$ [Klueter et al., 2006]) collected from the Heron Island reef flat.

Despite the ongoing reef-wide bleaching event and measured decline in coral endosymbiont densities, we find that NEP and NEC at both Lagoon sites did not significantly differ from estimates during the pre-bleaching period or prior estimates on other Great Barrier Reef lagoon communities of similar coral cover (e.g., 10 – 20 mmol $CaCO_3$ $m^{-2}$ $h^{-1}$: Albright et al., 2015; Pisapia et al., 2019; Stoltenberg et al., 2021). The lack of a bleaching effect was also mirrored in the slack water NEP and NEC data, which represented a much larger cross section of the lagoon community (~ 2 – 3 km transects), where bleaching was also observed (but not quantified during this study period). Importantly, these trends

differ from those observed by Courtney et al., (2018) during a 2015 bleaching event in Kaneohe Bay, Hawai`i (~ 10 % total cover), where a similar ~ 1 °C increase in mean reef temperature resulted in bleaching of 46 % of the coral community and both NEP and NEC were driven to zero. However, our results support those of Kayanne et al., (2005), where NEC and NEP remained relatively constant during a bleaching event (29 °C; 51 % bleached) in September of 1998 at Shiraho reef in Japan (5 – 7 % total coral cover). The critical difference between these studies is likely due to a threshold in total coral cover, where bleaching is less impactful on NEC when coral is not the dominant calcifying organism relative to the other calcifying constituents (sediments, rubble, calcifying algae, and other sessile or mobile gastropods and echinoderms) that are also known to contribute to the total reef carbonate budget and, in some cases, exhibit positive temperature-calcification relationships (Cornwall et al., 2019).

## 4.2 Estimated Organism Contribution to NEC at Elevated Temperatures

Importantly, if we consider that rubble observed in the Lagoon sites 1 and 2 (approximate cover of 4 %) was predominately covered in crustose coralline algae (CCA) and combine these with the other sessile calcifiers observed (that were predominantly *Halimeda* spp.; 3 % cover), then hermatypic corals were not the dominant reef calcifier. Further, if 60 % of the total coral cover was calcifying roughly 60 % slower due to bleaching (D'Olivo & McCulloch, 2017), this would imply that active calcifying coral cover was likely reduced to only 2 – 4 %. This adjusted 'calcifying percent coral cover' is minor compared to the sum of all other benthic constituents that were actively calcifying regardless of the SST conditions (Sediment + CCA + Halimeda = 72 %).

One possible explanation for the lack of any observed changes in NEC could be due to the simultaneous thermal enhancement of calcification in other benthic members when the reef seawater was warmed from 28.0 °C to 29.1 °C. To investigate the relative contribution to overall NEC from the assemblage

of benthic calcifiers at these respective temperatures, we created an equation based on reported rates
in the literature at 28.0 °C and 29.1 °C (Equation 1) where the summed community-level calcification
rate (NEC) at the respective temperature (T) is equal to the sum of the described calcification rates for
each benthic organism category (Net Organism Calcification: NOC) multiplied by the recorded cover
(Cover) across Lagoon sites 1 and 2 at that temperature (T).
$$Equation\ 1: NEC_T = \sum (NOC_T \times Cover_T)$$
To estimate the potential effect of a +1.1 °C change in seawater temperature on coral calcification for
corals observed within the lagoon study sites the following aquaria manipulation studies were
reviewed: Edmunds, 2005; Anthony et al., 2008; Cantin et al., 2010; Comeau et al., 2013, 2016; and
the following meta-analysis and modeling studies were reviewed: Lough and Barnes, 2000; McNeil et
al., 2004; Evenhuis et al., 2015; Kornder et al., 2018; Bove et al., 2020. Together, these studies suggest
mean calcification rates across coral genera most common to the Heron reef flat (*Acropora* spp.,
*Montipora* spp., *Porites* spp., *Pocillopora* spp.) at 28.0 °C ($4.53 \pm 2.31$ mmol CaCO$_3$ m$^{-2}$ h$^{-1}$) increase
by approximately 22 % when warmed to a temperature of 29.1 °C.  It is important to note this %
increase is highly variable and species specific, so numbers used here are simply for the purpose of
discussion. In comparison, calcification by crustose coralline algae (CCA), which is the next most
studied organism (see meta-analysis by Cornwall et al., (2019)), has not exhibited changes until
temperatures are as high as 5 °C above ambient temperatures. Therefore, no change was estimated for
mean reported rates ($0.36 \pm 0.09$ mmol CaCO$_3$ m$^{-2}$ h$^{-1}$) for commonly studied CCA species
(*Lithophyllum kotschyanum* and *Hydrolithon onkodes*).
Responses in calcification to warming for Halimeda algae are equivocal (Campbell et al., 2016; Wei
et al., 2020). If constrained to species commonly identified on the Great Barrier Reef (such as *H.*
*opuntia* and *H. cylindracea*; Aims, 2020) then it can be expected that increasing temperatures will
increase rates of calcification up to temperatures of 30 °C, above that they bleach and exhibit a negative
calcification response. As such, narrowed within the ranges observed during this study, calcification
rates of Halimeda ($3.33 \pm 2.29$ mmol $CaCO_3$ m$^{-2}$ h$^{-1}$) are estimated to increase by approximately 7.9
% in response to warming from 28.0 °C to 29.1 °C. Calcification responses to warming in carbonate
sediments are overall the least studied of the benthic categories in this study, but potentially the most
significant given the dominant cover of sediment . A study within the Heron Island lagoon indicates
daytime sediment calcification at 28 °C ($1.41 \pm 0.29$ mmol $CaCO_3$ m$^{-2}$ h$^{-1}$) would increase ~ 9 % when
seawater is warmed to 29.1 °C (Lantz et al., 2017).
When these trends are summed together with the expected 60 % decline in calcification for the
proportion of coral that was bleached, a collective 9.8 % decline in NEC can be expected (Fig. 4).
However, when each category is adjusted for the percent cover observed at the end of the study at 29.1
°C across both Lagoon sites, the total change in NEC increases by ~ 0.8 %. This is largely owed to
positive trends in the calcification of other benthic community members and provides an explanation
why no significant differences were observed in NEC during reef-wide coral bleaching. These
estimates illustrate how the decline in coral calcification may be overshadowed by thermal acceleration
in calcification in ahermatypic benthic calcifiers. Although some of these calcifiers still accrete
limestone structure (e.g., coralline algae), none replace the complex, three-dimensional structure
uniquely created by corals. Our findings highlight the need to better adjust how NEC is applied as a
metric for community function during bleaching events, as these data suggest warming may create a
divergence between estimated daytime NEC and actual reef growth on future degraded reef
ecosystems.
**4.3 Future Considerations**
Our study highlights three considerations that may affect NEC and need to be further investigated to
resolve monitoring issues for degraded coral reef communities. Firstly, the impact of night-time
dissolution on overall 24-h NEC. Estimates of NEC at night (n = 3) in the current study did not exhibit
a response to bleaching, but a higher frequency is needed. Courtney et al., (2018) hypothesized that
the dissolution signal was a major driver of the net 24-h zero NEC signal during bleaching. These
findings were more recently corroborated at the organism level by Orte et al. (2021), where algal turfs
on dead coral calcified at the same rate as coral during the day but transitioned to net dissolving at
night. This is supported by calcification responses to warming in the sediment, the most dominant
benthic member in this study, where warming-driven daytime increases in NEC were largely
overshadowed by night-time increases in dissolution (Lantz et al., 2017) and the sediments transitioned
to net dissolving over the full 24 h. These results suggest that future studies need include nighttime
measurements of NEC and NOC but also highlights the limitation of flow-metabolism approaches as
a representation of reef health given that not all reefs are easily accessible at night for such
measurements.
Secondly the longer-term changes in NEC (when bleached coral eventually dies or the thermal benefits
to other calcifiers expire) needs to be investigated if we are to accurately estimate community function
in future reef scenarios. In the current study we did not monitor the response in NEC following the
2020 bleaching event when a return to 28 °C or lower would likely reduce the thermal benefits to
daytime calcification in the sediment, rubble, live coral, and Halimeda algae that potentially masked
the minimized contribution from bleached coral. Under these assumptions, a 7.6 % decline in NEC
would be expected when temperatures return to 28 °C. Additionally, if we assume the bleached coral
eventually dies, and a 60 % reduction to calcification increases to a 100 % reduction, then community
NEC would in theory exhibit a total 13.1 % decline. These post-bleaching estimates may explain the
differences between this study and post-bleaching NEC estimates reported similarly degraded reef
transects at Lizard Island, Australia (3 % coral cover) by  McMahon et al., 2019, where post-bleaching
NEC in 2016 declined by 40 – 46 % relative to pre-bleaching estimates in 2008 when coral cover was
higher (~ 8 % coral).
Finally, the indirect feedbacks on NOC from non-calcifying community members (e.g., algae) and the
carbonate substrate they occupy also needs to be considered to predict future reef growth (Orte et al.,
2021). The sum of adjusted NOC (Fig. 4; 1.30 mmol $CaCO_3$ $m^{-2}$ $h^{-1}$) only explains 10.6 % of the
measured NEC (12.3 mmol $CaCO_3$ $m^{-2}$ $h^{-1}$). Such discrepancies may be explained the exclusion of the
21 % of space occupied by non-calcifying algae in the NOC summation exercise in Fig. 4. It is possible
algae can provide positive feedback mechanisms to coral calcification through adjacent algal-driven
NEP (and subsequent modifications to the surrounding seawater carbonate chemistry; Gattuso et al.,
1998; Unsworth et al., 2012) or the endolithic micro-calcifiers living inside the dead carbonate
substrate colonized by algal communities (Orte et al., 2021). For example, endolithic microflora
(Cyanophyta and Chlorophyta) living within carbonate rocks have been found to modify interstitial
pH just beneath substrate surface to values as high as 8.5 (Reyes-Nivia et al., 2013), thereby creating
localized zones supersaturated with aqueous $Ca^{2+}$ and $CO_3^{2-}$ ions (Krause et al. 2019) and promoting
the inorganic precipitation of minerals such as brucite, micrite and dolomite. Critically, these
microfloral communities are more diverse and abundant when living beneath turf algae compared to
corals (Gutierrez-Isaza et al. 2015), are comparable in their productivity to overlying turf algae
(Tribollet et al. 2006), and have been found to precipitate dolomite at an accelerated rate when seawater
temperatures were increased from 28 °C to 30 °C (Diaz-Pulido et al. 2014). Taken together, this shows
that these microfloral communities have the capacity to influence bulk seawater chemistry
measurements particularly during coral bleaching events, where warm and well-lit conditions promote
their growth. In addition to these microflora, various cryptic infaunal and endolithic macrofauna
calcify to produce protective shells or burrows (e.g., Diaz-Castaneda et al., 2019) and may also be
contributing to NEC signal measured during the bleaching event.
**4.4 Conclusions**
Ocean warming, and subsequent coral bleaching events, have already degraded coral reef ecosystems
for over four decades and will continue to degrade coral reefs worldwide, reducing their capacity to
provide complex, three-dimensional habitat structure. While estimates of NEC via the alkalinity
anomaly technique may be an appropriate benchmark of community function well after bleaching

events have occurred and degradation to the coral community is fully realized, the results from this study highlight the shortcomings of using this approach to estimate daytime NEC when monitoring the effect of bleaching on reef accretion in real-time. These results, in conjunction with available literature on the importance of nighttime dissolution, suggest that flow-metabolism approaches to estimate community health may be limited to reefs accessible at night (e.g., those near a research station or without navigational hazards). Moreover, our study highlights that if coral cover continues to decline as predicted, NEC may no longer be an appropriate proxy for reef accretion as the proportion of the NEC signal owed to ahermatypic calcification increases. Additional estimates of NEC during bleaching events are urgently needed to further explore the potential decoupling of positive NEC and reef growth. Concerningly, the data herein suggest that NEC may begin to exhibit limitations as monitoring tool for reef growth when coral becomes the minority benthic constituent.

**Author Contributions**

Coulson Lantz is responsible for study design, data collection and analysis, and writing. William Leggat is responsible for study design, data analysis, and writing. Jessica Bergman is responsible for data collection, analysis, and writing. Alexander Fordyce is responsible for data collection, analysis, and writing. Charlotte Page is responsible for data collection, analysis, and writing. Thomas Mesaglio is responsible for data collection and analysis, and writing. Tracy Ainsworth is responsible for study design, data analysis, and writing.

**Acknowledgements**

This work was funded by the Australian Research Council DP 180103199. We thank the Heron Island Research Station scientific staff for their support during research. We also thank all iNaturalist users that helped identify the invertebrates photographed during this study, especially Joe Rowlett, Sean Ono, Frédéric Ducarme and Pierre Mascar.

**Data availability statement**

Data is presently being submitted to PANGAEA data repository and a DOI will be provided upon
completion.

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

**Tables**

**Table 1**: Percent cover (Mean ± SD) measured during point-contact and photo-quadrat surveys. Data for point contact surveys were pooled across triplicate transects and repeated survey efforts (n = 6 site$^{-1}$) within each Lagoon site area. Data for photo-quadrat surveys were pooled across triplicate transects and repeated survey efforts within each Lagoon site area (n = 360 site$^{-1}$).

| Category | Lagoon site 1 | | Lagoon site 2 | | Total |
|---|---|---|---|---|---|
| | **Point Contact** | **Photo Quad** | **Point Contact** | **Photo Quad** | **Mean Cover** |
| Hard Coral | 3 ± 2 % | 3 ± 2 % | 8 ± 3 % | 9 ± 3 % | 6 % |
| Soft Coral | 1 % < | 1 % < | 1 % < | 1 % < | 1 % < |
| Algae | 27 ± 4 % | 18 ± 5 % | 23 ± 4 % | 16 ± 4 % | 21 % |
| Other Calcifier | 3 ± 2 % | 2 ± 2 % | 6 ± 1 % | 2 ± 2 % | 3 % |
| Rubble | 4 ± 3 % | 2 ± 2 % | 5 ± 3 % | 3 ± 3 % | 4 % |
| Sediment | 62 ± 6 % | 74 ± 7 % | 57 ± 7 % | 69 ± 6 % | 65 % |

**Table 2**: Mean values for physiochemical parameters measured at Lagoon site 1 and Lagoon site 2
over the course of the study. Temperature and light were logged continuously at 15-min intervals.
Temperature data are separated by the pre-bleaching period (Jan 22 – Feb 1 2020) and bleaching period
(Feb 2 – Feb 10 2020). Salinity was measured with each collected water sample (n = 60 site$^{-1}$). Depth
was measured at peak low tide at 5m intervals along each transect (n = 120 site$^{-1}$). The flow meter was
rotated between downstream water sample collection locations on each day of collection (n = 5 site$^{-1}$).

| Parameter | Lagoon site 1 | Lagoon site 2 | Mean |
|---|---|---|---|
| Temperature (° C) Pre-Bleaching | 28.1 ± 1.3 | 28.0 ± 1.3 | 28.0 ± 1.3 |
| Temperature (° C) Bleaching | 29.0 ± 1.5 | 29.1 ± 1.5 | 29.1 ± 1.5 |
| Salinity (PSU) | 35.6 ± 0.2 | 35.7 ± 0.2 | 35.7 ± 0.2 |
| Light (μmol m$^{-2}$ s$^{-1}$) | 328 ± 247 | 336 ± 254 | 332 ± 251 |
| Depth (cm) | 37 ± 7 | 36 ± 6 | 37 ± 7 |
| Flow (cm s$^{-1}$) | 21.6 ± 2.9 | 19.2 ± 3.8 | 20.4 ± 3.3 |

**Table 3**: Change in the relative percent area (Mean $\pm$ SD) of coral tissue exhibiting paling or bleaching
(Bleached Coral Tissue) and relative percent area (Mean $\pm$ SD) of sediment exhibiting overgrowth in
the form of visible cyanobacteria mats or Chlorophyta growth (Overgrowth on Sediment) over the
course of three different survey efforts. Data for each date are pooled across parallel transects within
each Lagoon site ($n = 120$ site$^{-1}$).

| | Study Site | Jan 24 2020 | Feb 6 2020 | Feb 12 2020 |
|---|---|---|---|---|
| **Bleached** | Lagoon site 1 | $0 \pm 0$ % | $16 \pm 3$ % | $55 \pm 8$ % |
| **Coral Tissue** | Lagoon site 2 | $0 \pm 0$ % | $24 \pm 6$ % | $65 \pm 10$ % |
| | | | | |
| **Overgrowth** | Lagoon site 1 | $2 \pm 1$ % | $4 \pm 2$ % | $10 \pm 2$ % |
| **On Sediment** | Lagoon site 2 | $3 \pm 1$ % | $5 \pm 3$ % | $14 \pm 5$ % |

**Table 4**: Mean ± SD values for daytime net ecosystem production (NEP; mmol $O_2$ $m^{-2}$ $h^{-1}$) and net
ecosystem calcification (NEC; mmol $CaCO_3$ $m^{-2}$ $h^{-1}$) for Lagoon site 1 and Lagoon site 2, where the
Eulerian approach was used (n = 12). NEC for the slack-water approach included for daytime (n = 11)
and night time (n = 3) estimates. Data are separated by the pre-bleaching period (Jan 22 – Feb 1 2020)
and bleaching period (Feb 2 – Feb 10 2020; n = 8). Nighttime rates for NEC are included NEP values
are not included for the slack-water approach given the large source of error in air-sea oxygen
exchange.

| Approach | NEP (mmol $O_2$ $m^{-2}$ $h^{-1}$) | | NEC (mmol $CaCO_3$ $m^{-2}$ $h^{-1}$) | |
|---|---|---|---|---|
| | Pre-Bleaching | Bleaching | Pre-Bleaching | Bleaching Period |
| **Lagoon site 1** | 35.0 ± 12.7 | 39.7 ± 9.6 | 12.5 ± 4.5 | 12.6 ± 4.8 |
| **Lagoon site 2** | 44.4 ± 13.6 | 38.7 ± 13.8 | 13.3 ± 5.7 | 12.3 ± 5.4 |
| **Slack Water (day)** | | | 11.0 ± 2.9 | 10.5 ± 3.0 |
| **Slack Water (night)** | | | - 2.8 ± 0.7 | - 3.4 ± 1.3 |

## Figures

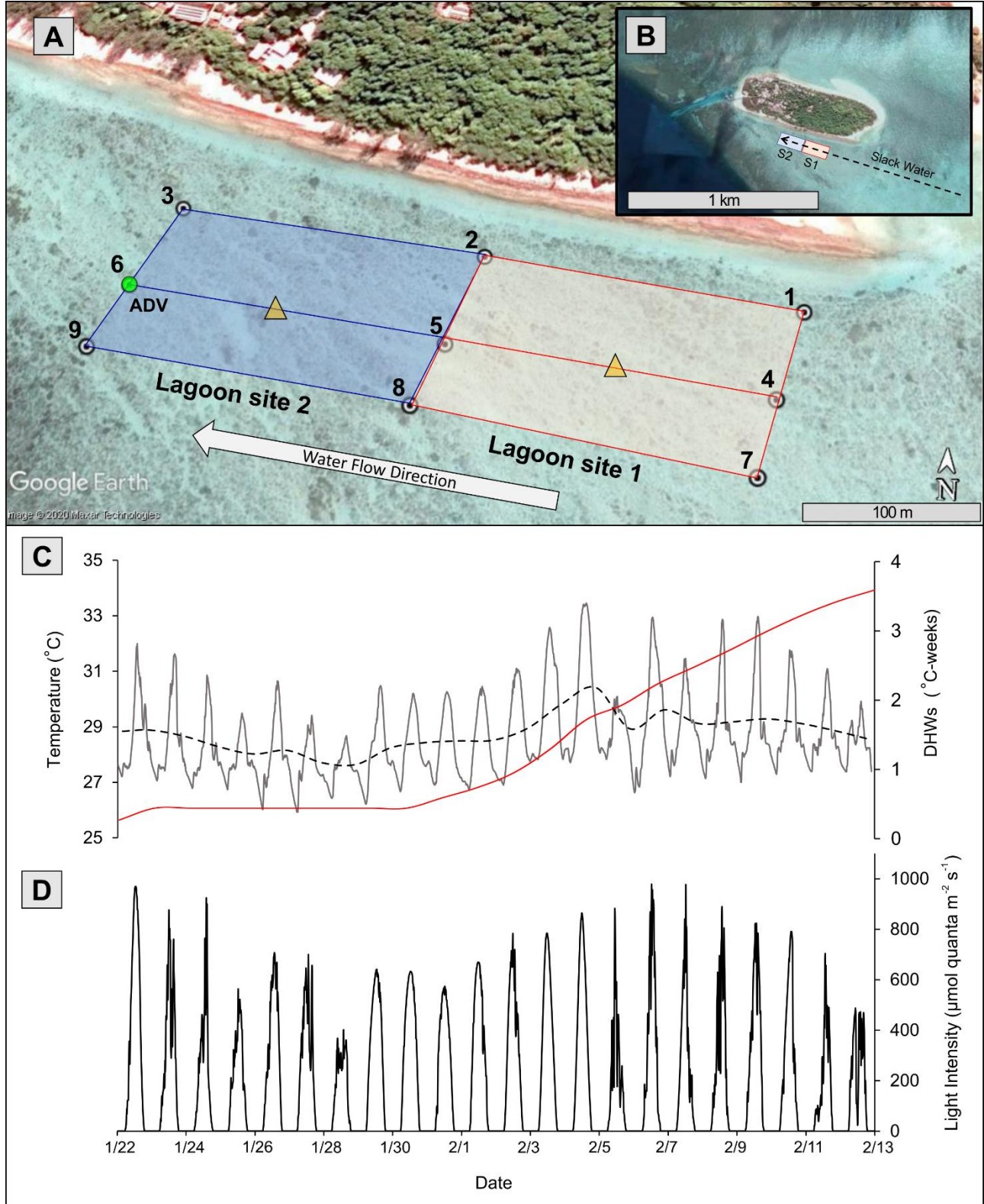

**Figure 1**: A) Study area (100 m scale) subdivided into Lagoon site 1 (red) and Lagoon site 2 (blue),.

White numbered circles (1 – 9) indicate of location water samples and temperature loggers. Yellow

triangles indicate location of light loggers.  B) Study area (1 km scale) showing Lagoon site 1 (S1) and
Lagoon site 2 (S2) in relation to Heron Island and the larger slack-water area. C) In-situ lagoon
temperature (°C) averaged across both sites measured by temperature loggers. Black dashed line
represents the 24-h average of these temperature data and red line indicates the accumulation of degree
heating weeks (DHWs; °C-weeks) in these data. D) Light intensity (μmol quanta $m^{-2} s^{-1}$) averaged
across two light loggers. Green circle represents location of ADV flow meter during Eulerian
estimates. All data were recorded at 15-min intervals from Jan 22 to Feb 13 2020. Aerial photograph
is provided by © Google Earth.

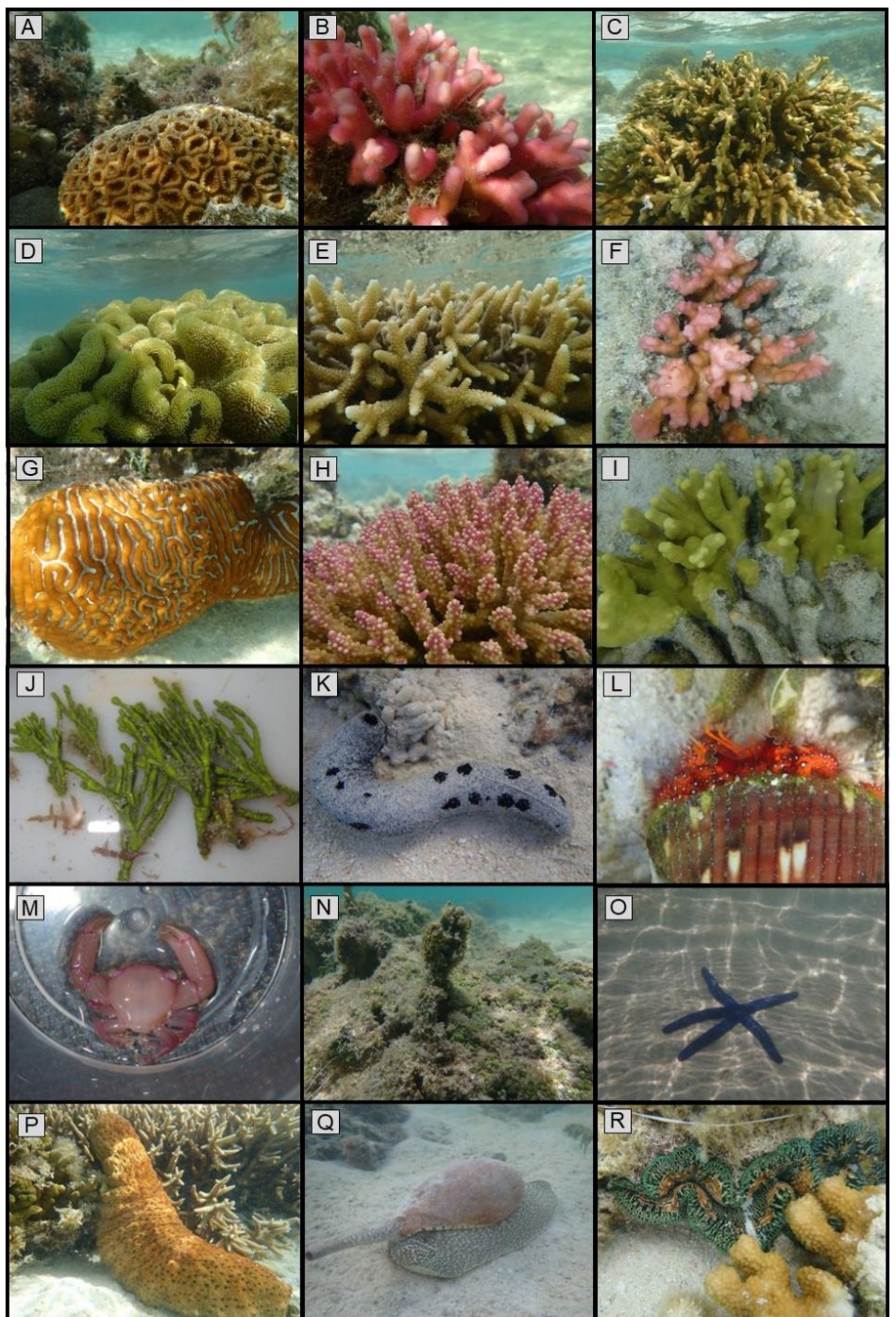

**Figure 2**: Cross-section of coral, algal, and invertebrate diversity observed within the study area. A) *Dipsastraea* sp.; B) *Stylophora pistillata*; C) *Montipora digitata*; D) *Sarcophyton* sp.; E) *Acropora* sp.; F) *Pocillopora sp.* G) *Platygyra* sp.; H) *Acropora secale*; I) *Porites attenuata*. J) *Halimeda* sp.; K) *Holothuria atra*; L) *Dardanus megistos*; M) *Trapezia serenei*; N) Assemblage of *Caulerpa* sp. and *Laurencia* sp. algae covered in scum sp.; O) *Linckia laevigata*; P) *Stichopus herrmanni*; Q) *Melo amphora*; R) *Tridacna maxima*.

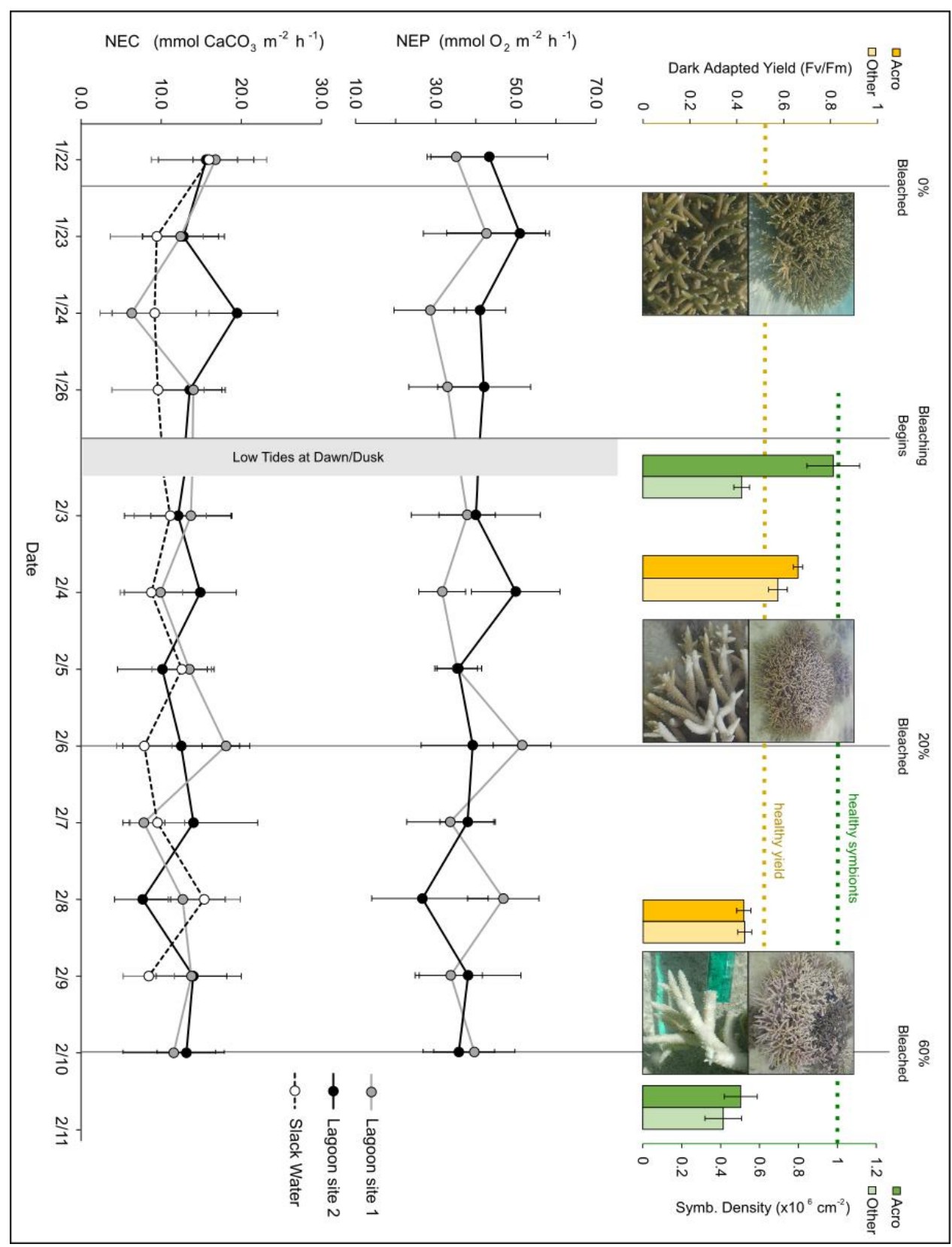


**Figure 3**: Dark adapted yield (yellow; top left), Symbiodiniaceae densities (green; top right),

Rates of net ecosystem production (NEP; middle) and net ecosystem calcification (NEC;

bottom) in at Lagoon site 1 (grey), Lagoon site 2 (black), and the larger reef area (Dashed;

slack water). Dashed yellow and green lines indicate expected healthy values for dark adapted
yield and Symbiodiniaceae densities, respectively. Grey vertical lines indicate the date of
photo-quadrat surveys and the resulting percent area of coral that was bleached. NEP and NEC
estimates were paused between Jan 26 to Feb 3 due to low tides occurring at dawn and dusk in
low light conditions, preventing estimates of NEC. Slack-water estimates are excluded from
the NEP data given the large error associated with air-sea gas exchange corrections.

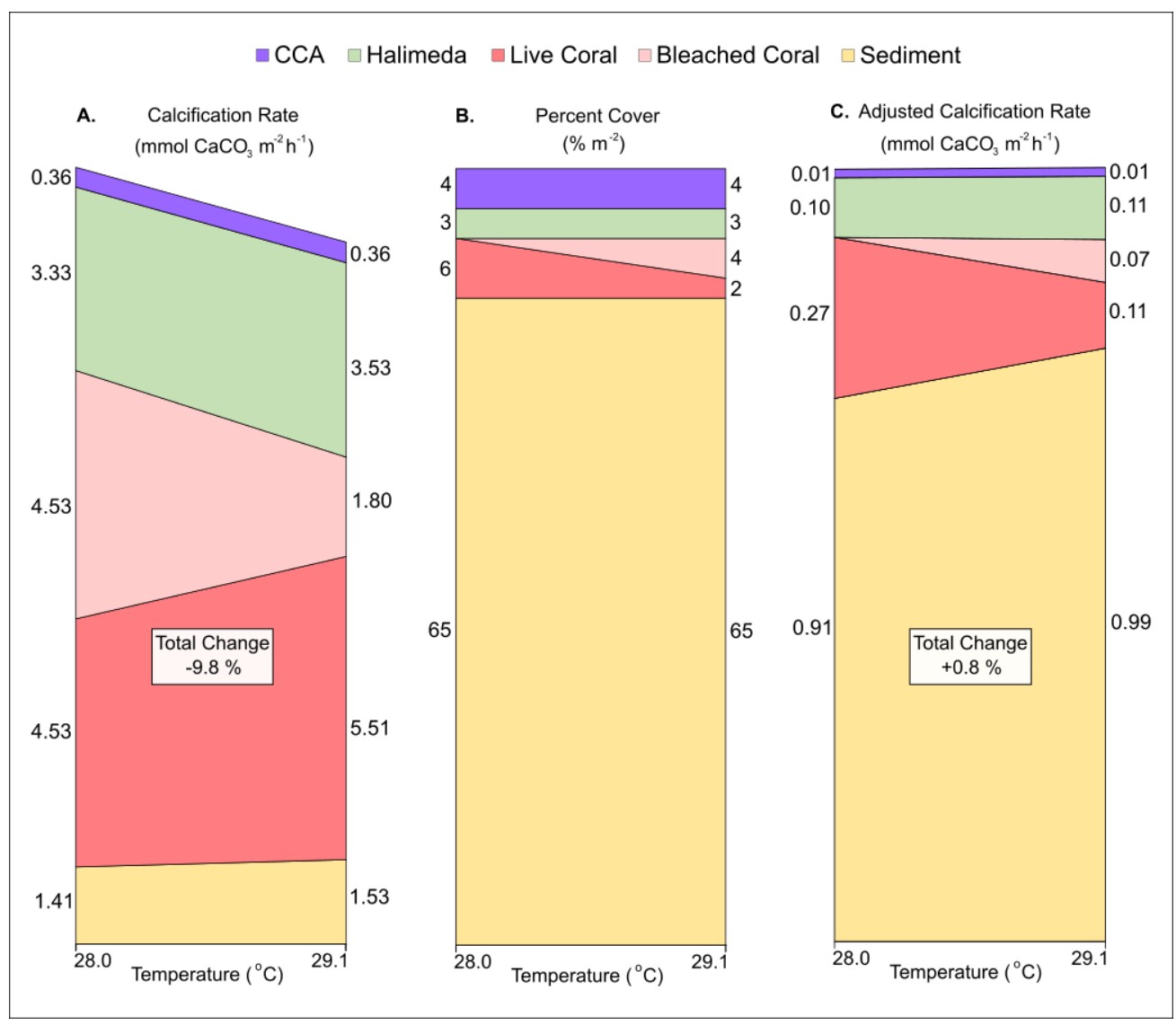


**Figure 4**: Visualisation of the changes caused by a transition from pre-bleaching (28.0 °C) to bleaching (29.1 °C) temperatures in A) estimated individual organism calcification rates from the literature (converted to mmol $CaCO_3$ $m^{-2}$ $h^{-1}$), B) percent cover across Lagoon site 1 and Lagoon site 2 combined, and C) the "adjusted the calcification rate" (mmol $CaCO_3$ $m^{-2}$ $h^{-1}$) calculated by multiplying A. x B. at each temperature. Total change (%) represents the percent difference in the sum of all rates at 29.1 °C relative to 28 °C. Rubble and Other Calcifier categories were assumed to be CCA and *Halimeda* spp., respectively.