# Peer review of "Will daytime community calcification reflect reef accretion on"

_Biogeosciences, 2021_

## Author Comment (AC1)

**Reviewer Comment #1**

This study presents an interesting idea that temperature-induced increases in metabolism of various non-coral calcifying organisms could offset the expected effects on coral calcification during bleaching events, and thus that ecosystem-based measurements of calcification may not fully represent disturbance to these degraded habitats. The data are interesting and are presented in this new and insightful light, but the manuscript needs clarification in many aspects, and especially further consideration of statistical power.

Methods: not enough detail is provided to understand the methods. The supplement helps, but is far too long in my view (the supplement itself is separated into the main sections normally in a paper (Methods, Results, Discussion), so it almost reads like a separate paper. Importantly, essential details to understand the study are in the supplement. For example, the description of the Eulerian approach to NEC and NEP is too brief in the main text. The reader needs to access the supplement to begin to understand what was done here. Additionally, even the supplement is lacking some details, like precisely which samples were used for upstream and downstream TA and DO, why there is a 3600/100 in the equations, how was residence time calculated (e.g., where was the current meter?), how was TA titrated, etc.

**We thank Reviewer #1 for their comments. We understand the reviewer would like more of the methods in the main text and we oblige. We can provide details of the placement of the current meter and more specific information on sample collection and alkalinity analysis. The core citation for these methods (Langdon et al., 2010) details why 3600/100 is in the equations (to convert cm s-1 to m h-1). We will endeavour to include as much of this information as possible in the main text.**

**We would also clarify that thermally accelerated calcification is not the main conclusion of this paper but rather a proposed explanation, among 2 others (algal calcification, nighttime dissolution). Since the submission of this paper a publication by De Orte et al., 2021 (Unexpected role of communities colonizing dead coral substrate in the calcification of coral reefs, 2021, L&O) has provided compelling evidence for our other proposed mechanism: inorganic precipitation in dead coral substrate caused by localised algal photosynthesis. We plan to add this citation and more back of the envelope calculations (similar to Figure 4) which mathematically explain how the domination of fleshy algae in the lagoon growing on dead coral may have masked the decline in coral calcification. Finally we will also be adding more to nighttime dissolution aspect given other recent work.**

**Overall, the main conclusion of this study is that 2 traditional flow metabolism approaches (Eulerian and Slack Water) did not discern differences in daytime calcification due to bleaching. We then propose 3 mechanisms in the discussion based on the available literature: thermally-accelerated calcification, algal-dead coral calcification, and increased nighttime dissolution.**

Results/Discussion: given the relatively high variance and large error bars on the NEC rates, is it surprising that there were not significant differences observed before/after bleaching, especially given the low number of independent samples (days)? An assessment of statistical power would be highly useful. Of course, it shouldn't be concluded that bleaching doesn't affect NEC. Rather, this study did not reject the null hypothesis that bleaching has no effect on NEC. It's a key distinction, one that is glossed over and somewhat misinterpreted here.

**Perhaps these concerns will benefit from moving more methods from the SI to the main text. There were triplicate transects for 2 different reef zones measured daily. We understand the key distinction between not rejecting the null hypothesis and will endeavour to use more statistical approaches to clarify this point. We will also employ more statistical power to show there were no significant differences (or elucidate if there are in fact are).**

Throughout: more clarity needed in how quantities were calculated and exactly how each type of data was used. For example, the text section 3.2 describes satellite SST begin to accumulate heat stress in February and refers to Figure 1. But from what I understand of the caption of Figure 1, only the in situ logger data are shown in that figure.

**This confusion is understood. Figure 1 displays the in-situ logger data (black line) and the accumulation of heat stress in these data (red line). The text discusses that accumulation of heat stress in the satellite data was the overall driver to begin this study. This can clarified.**

Need to describe statistical approach in main text.

**Supplemental material discussion of statistical methods will be moved to the main text.**

Throughout: it seems odd to say "community NEC" — usually, it's either "net community calcification" or "net ecosystem calcification"

**We understand this point. The word "community" is used to help differentiate from organism-level calcification. We can change the nomenclature to net ecosystem calcification and net organism calcification.**

Throughout: need to decide if there is a space between numbers and % symbols or not.

**Will edit and add a space between numbers and %.**

Introduction: the discussion of existing literature is good and thorough, but perhaps there should be clearer differentiation between the effects of ongoing bleaching vs bleaching-induced mortality. Currently, the text describes these similarly, but it seems likely there would be different NEC responses to bleached (but living) corals as opposed to dead corals.

Line 88: Didn't Kayanne also observe a decline in NEC after bleaching in Palau though?

**This is correct, they found differing responses at different reefs. We recognize this reviewer's work and others earlier in the introduction where we established that the expected response is a decline in NEC in response to bleaching (L54 – 65). We can add the Kayanne Palau results to this evidence as well. The purpose of the text in L88 is to highlight examples in the literature that have curiously not shown a decline in NEC.**

Lines 88-89: again, bleaching vs bleaching-induced mortality seem to be conflated. Kayanne describe changes after bleaching-induced mortality, which the present text is comparing to NEC during a bleaching event (but with still-living corals). More clarity is needed about the difference between the two.

**We can add into the text the words "after" instead of "during" for the bleaching event. We understand the reviewer's point of conflating NEC during bleaching vs. after bleaching, but studies are so limited of either situation that we instead use the introduction to simply introduce any and all studies with NEC and bleaching through the lens of described coral cover. This critical difference between during bleaching vs. after bleaching is discussed on L330 in the context of the results here. We will endeavour to clarify better measurements that are during or after bleaching events.**

Line 134: should state how many points were used on each image, and if the points were randomized.

**100 points, not randomized, its a standard grid. This will be added to text.**

Line 152: "using using"

Line 232: delete "extremely"

**Noted will change these typos. Thank you**

Figure 4: why not just have 2 bars for each temperature?

**The figure is shown as one bar to illustrate the relative change in each benthic category as temperatures increased.**

---

## Author Comment (AC2)

**Reviewer Comment #2**

**General Comments**

Overall, the theory of the study is great, and inter-disciplinary work like this is great to see. However, the linking of study elements (for example, quantification of the relationship between Symbiodiniaceae densities or photosynthetic yield and NEC) could be more deeply explored. Importantly, I find the methodology lacking necessary information to determine the validity of the results and many facets of the methodology and further analyses require justification. In the supplementary material, the equations used to calculate metabolic rates are not well defined and in their current state may be incorrect. The authors should take care with the accuracy of information presented from the literature and with the appropriateness of citations to fit the manuscript's narrative. I find the results interesting, but their main point seems oversold and broadly declarative with the data that is presented.

**We thank Reviewer #2 for their comments. Similar to other reviewer comments, moving methods text from the supplemental material to the main text will help address these concerns. If the reviewer is referring to the thermally-accelerated calcification hypothesis as the "main point", we would invite the editor to see our explanation provided for reviewer #1 clarifying this is not the main conclusion of the paper but 1 of 3 proposed hypothesis to stimulate thought around this subject.**

 **Specific Comments**

Line 67-69 this information is incorrect. The bleaching event year is 2016, and the 2016 survey by Pisapia et al. cited here occurred after the bleaching event occurred in 2016. The 2018 survey showed an increase in NEC compared to 2016, but was still depressed related to pre-disturbance rates (see Abstract and Table 1 in cited paper).

**Thank you for catching this. This is a typo, the text should read 2016. This text is straddling two studies (Pisapia et al., 2019 and McMahon et al., 2019).**

I disagree with the sentence from Line 74-76: A community transitioning to algal dominated from a coral dominated community would likely demonstrate changes in NEC and NEP, as indicated by many prior studies. The citation here (Courtney et al., 2018) is inappropriate because this paper does not indicate that NEC will become less effective with reef state transitioning. Rather, the authors state ". . . bleached coral reefs that recover quickly likely experience ephemeral reductions in reef NEC while systems shifting to alternative non coral-dominated states are likely to face lasting decreases in NEC." Reduced or low NEC, regardless of the cause or the dominant benthic class, is what is useful when investigating reef state.

**The purpose of this text is to explain that if anthropogenic stressors continue to increase the ratio of algae:coral on reefs, then NEC may no longer be representing reef accretion by hermatypic coral. This idea is proposed by Courtney et al. 2018 in the discussion:**

*In contrast, measurements of NEC at Shiraho Reef, Japan did not change during the September 1998 bleaching event, where 51% of the total 7.1% total coral cover was bleached compared to a recovery survey conducted in September 1999 with 6.7% total coral cover and no bleaching observed (Kayanne et al. 2005). Kayanne et al. (2005) hypothesized*

*that calcification by living bleached corals, calcifying algae, and benthic foraminifera may have compensated for bleaching-induced losses in NEC at Shiraho Reef. Indeed, the dominant calcifiers of coral reefs include corals, red coralline algae, molluscs, green calcifying algae, and benthic foraminifera (Montaggioni and Braithwaite 2009), but their relative contributions to coral reef CaCO3 budgets and how these change under different reef states are uncertain.* ==*This raises the question and need to further quantify the relative importance of contributions by other calcifiers to coral reef NEC especially for low coral cover (< 10%) and bleached coral reefs.*==

Line 84-85, this statement is incorrect. The Kayanne et al. 2005 abstract states 'All the metabolic parameters, Pg, R, E and calcification (G) were reduced by half after the bleaching,', and no pre-bleaching rates were estimated.

**In addition to the text above from Courtney et al., 2018 summarizing Kayanne et al., 2005 Shiraho Reef results, there is also specific text in the Kayanne paper which explains differing results between Palau and Japan. Yes, the abstract states G was reduced by half but that is for Palau. Within the actual text they found differing responses in Palau and Shiraho reef at Ishigaki Island, Japan. The following is taken specifically from Kayanne et al., 2005 section 3.1.4:**

*However, E was lower in September 1998 (36 mmol m_2 d_1); this was during the bleaching period, when the mean coverage of living coral was 7.1% (*==*51% of which was bleached*==*, so that the coverage of corals with symbiotic algae was 3.6%). After the bleaching, E increased over the course of recovery. By contrast, G (calcification)* ==*remained nearly constant during the September 1998 bleaching at Ishigaki Island.*==

**We fully admit starting on L56 that the normal expectation is for bleaching-induced decline in NEC. We can add Kayanne's Palau results to further support this expectation. When we found no change in NEC, we performed an exhaustive literature review to find any possible examples where this was also observed. We fully admit this should not be the norm and present the information specifically in this order.**

Methods:

Overall, I found the methods do not provide enough information or justification to assess the validity of the rates or to undertake a follow-up survey. One major concern is that the time window stated (1100-1500) is not representative of calcification over a diel cycle (which is what is typically used when discussing reef state). Different stressors can affect calcification/dissolution differently depending on the time of day. Prior studies have shown that impacted corals show a significantly higher rate of low-light dissolution than non-impacted corals, even when daytime or peak sunlight calcification is not affected.

**Enough information is provided in the supplemental material to conduct follow up measurements and this can be easily rectified with more information moved to the main text.**

**We agree that calcification can differ depending on time of day and light. Perhaps this needs to be further clarified, but measurements on the Heron reef flat can only be conducted during a 3 hour period around low tide. Outside this period, there is too much water on the reef and mixing with the offshore water to measure adequate**

changes in water chemistry. Therefore, the tides must be followed and one cannot make multiple measurements on the same day.

**It is important to note that the purpose of this study was not to describe the diel calcification trends on Heron reef flat, this has already been done multiple times, most recently by Stoltenberg et al., 2020. The purpose was to compare pre-bleaching and during-bleaching NEC. Midday at full light provides the best opportunity measure peak community metabolism and ensure changes in seawater chemistry were large to reduce error. We fully admit in the discussion as a follow up more measurements need be conducted now at dawn, dusk, and night. Processing samples along with other coincidental work on the reef during the bleaching event prevented enough time in the day to do it all, especially nighttime measurements.**

Why were coral fragments gathered for PAM fluorometry rather than assessment in situ? More information is needed on PAM measurements. How were those fragments chosen, how many of 'bleached' and healthy? Was there a control for PAM measurements or Symbiodiniceae densities during the bleaching? Were pre-bleaching yield or Symbiodiniceae densities measured, it does not appear so in Fig. 3.

**More information can be provided on exactly how coral fragments were gathered. In short, the listed taxa were gathered across the reef during the bleaching event. We gathered live branches of all these taxa to see if they were healthy or not.**

**The model PAM listed is a benchtop unit, we did not have an underwater PAM, so corals could not be measured in-situ.**

**It is important to note that examples of a flow metabolism study setting up all of these type of control measurements before the bleaching begins is extremely rare simply because our predictive abilities are generally not dependable enough to mobilize an entire flow-metabolism team and gear. This is why most studies like McMahon et al., 2019, Courtney et al., 2018, Pisapia et al., 2019 measure either during or after.**

**This study did not begin with a certain expectation of bleaching, it was initially started as a study to relate community-level NEC and NEP to the census approach from benthic surveys. In the middle of this satellite data indicated accumulation of heat stress and signs of bleaching began. Realizing this opportunity, the study objectives were quickly shifted to take advantage of this event. We recognized that qualitative examples of bleaching (photo quadrats, white corals) may not be enough to prove the coral's health was compromised and decided some physiology measurements (PAM, symb. Densities) would be a great way to add strength to this statement. Perhaps some text would be beneficial to explain how this "opportunity" arose and why pre-bleaching physiology measurements are not available.**

Equation 1 & 2 in the Flow Respirometry Approach: I am making assumptions here because I don't understand these equations. If $u$ u is current speed (cm s$^{-1}$)(not stated), then in the context of these equations, you are multiplying current speed by 3600 to transform from per second to per hour and dividing by 100 to transform from cm to m. However, if I am correct, this means that you have divided out your length component (m/m). The residence time is the term that needs to be on the denominator to provide the unit of mmol/m$^2$/hr. If I am incorrect here, then more definition needs to be put into your equations to show that you are

calculating the correct values. Typically, residence times are calculated separately than metabolic values using transect length, current speed, and length/time averaged depth (See Supplementary Information in DeCarlo et al., 2017, or Davis et al., 2020).

**The questions used in this study, including why the equation has 3600/100, is detailed in Langdon et al., 2010. Since this seems to be a point of concern, we can add all of these details to the main text.**

'Slack water' sampling is not made under the correct conditions. Slack water indicates no (or very slow) water movement, so this may be an issue of semantics and you really mean Eulerian sampling (see Silverman et al., 2014 and McMahon et al., 2019 for examples and comparisons). More information is needed here. If you are looking at changing water chemistry in the same place, you need either 1) and end member (an initial value), or 2) More specific calculation of your depth-averaged residence time over space. Did you have a current meter? If so, where was it placed? Maybe add to site map?

**Yes, the slack-water name sounds like water isnt moving, but there are multiple studies which define that slack water simply means the water is contained in a basin and circulated within. This method has been used previously in the exact same location by Stoltenberg et al., 2020 (Late afternoon seasonal transition to dissolution in a coral reef: An early warning of a net dissolving ecosystem? GRL) and at nearby One-Tree island by Shaw et al., 2012 (Impacts of ocean acidification in naturally variable coral reef flat ecosystems, JGR). It has been traditionally employed on reef flats that are separated from the open ocean at low tide even if there is still water moving within the reef flat.**

**We have a site map (Fig. 1) and can move text from the supplemental material which indicates where the current meter was placed (at the end of the middle transect) for the eulerian measurements into the main text and make a note on the map in Fig. 1. The current meter was moved between the three end sample locations before the study began to ensure the current speeds were roughly the same 50m apart and this was corroborated with flourescein dye measurements across the transect area. This text is in the SI and can be moved to main text.**

'Slack water' Methods state: water samples were collected from the same three locations (n = 3 day-1) two hours before 84 peak low tide and one hour following." Where were these locations and at what interval were they taken? Were they taken at the same time/place as the 'flow' samples were taken?

**This is all detailed the supplemental material and we will move this text to the main document. The slack-water interval is 3 hours (two hours before, one following = 3 hour interval). The eulerian interval is as close in time as possible, as fast as we could walk 200m across the reef (upstream and downstream sample). The locations of all these sampling locations is noted in Fig. 1. This will be further clarified by which sites were specifically used for Eulerian and which for Slack-water, but in general the downstream samples for the eulerian approach is the same location of the slack-water samples (which are collected in the same location 3 hours apart).**

**We thought that the effort of using both the Eulerian and Slack-water approach would impress reviewers, as it is rare to compare both approaches simultaneously to provide strength to community metabolism estimates. Unfortunately, this seems to provide more**

**confusion, specifically with a misunderstanding of the slack-water approach. To prevent this with other readers, we will endeavour to be more explicit with this in the main text.**

Figure 4: More information is needed on where the calcification rates were taken from in the literature. How were differences in calcification rates determined for different specific temperatures? Is this what's described in L285 – 288? If so, is the 1.1 degree change in temperature determined from absolute differences in temperature or for corals which bleached after 1.1 degree increase? If the bleaching temperature was 29.1, the calcification rate indicated here at 29.1 degrees should represent calcification rates under bleaching. Please specify.

**We will take this as a point of emphasis to move the text into the main document so other readers do not do the same. Information on where calcification rates are taken is currently provided in the supp (S.7) and this will be moved to the main text with more detail.**

**Per L271-273, L306 in the main text and the description of Figure 4, the calcification rate for coral indicates the rate expected based on coral NEC at 29.1 x reduced calcification rate expected under bleaching x the amount of coral bleached on the reef.**

Lines 306–308: How does this 9.8% expected decline in NEC compare with your observed results?

**There was no significant changes in NEC (Table 4), so its a 9.8% larger decline than observed. We can add more text to L306 that the observed change was essentially 0%.**

L 312–316: This is a good description of this result, and it is an interesting result. However, I think the wording in the abstract and conclusions are too strong with the data provided to support the argument.

**We can reduce the strength of the conclusions in the abstract. Overall, the unobserved decline in NEC despite observed bleaching provided the opportunity to hypothesize 3 different drivers behind why this occured: 1) Thermally-accelerated calcification 2) Algal/Dead Coral calcification 3) Increased nighttime dissolution. On L312-316 we discuss the thermally accelerated calcification in unbleached sessile calcifiers. Since the submission of this manuscript, a publication by De Orte et al., 2021 (Unexpected role of communities colonizing dead coral substrate in the calcification of coral reefs, 2021, L&O) has provided compelling evidence for our other hypothesis outlined in L343 (dead corals with algae might be calcifying) and we will be using these new data to add to this section.**

**Overall, perhaps it is necessary to correct the text to clarify that the thermally-accelerated calcification is not a conclusion. Its simply an idea in the discussion to stimulate thinking about what could be driving daytime NEC during a bleaching event. We provide back of the envelope calculations and Figure 4 to flesh out this idea and as noted will now add more to flesh out the idea of algal calcification.**

**Technical Corrections**

Fig.7 is referenced a few times but no figure 7 is included, change to Fig. 4.

**Noted, thank you. Fig. 7 will be changed Fig. 4**

Information in the table 2 caption should be placed in the methods.

**Noted, this will be moved to the methods.**

---

## Author Comment (AC3)

**Reviewer Comment #3**

General comments:

This is a timey study that discusses the possible divergence between estimates of NEC and reef growth in degraded coral reefs. The authors provide an interesting perspective that thermal enhancement of calcification in other benthic members may highly influence NEC, especially in reefs where coral cover is low.

The limited amount of nighttime NEC measurement is a weakness of the study. Nighttime dissolution could significantly influence the 24 hours NEC signal, especially if other benthic groups are contributing substantially to the calcification signal. The authors do a good job discussing this issue in the "future considerations" section. However, the lack of these data could have influenced the conclusions of this study.

There are important information missing in the main text while the SI is too long. Authors could change the structure of the paper by including some sections of the SI (e.g. S2.2.) in the main manuscript.

Overall the paper is well written but there are some references and details missing which are highlighted in the specific comments.

**We thank reviewer #3 for their review of this manuscript. Their concerns fall in line with Reviewer #1 and #2, requiring more text from the SI to moved to the main text which can be easily rectified. Clarifications among the citations will also be fixed.**

**We also agree that nighttime dissolution could be a major driver of the NEC decline that we did not measure and, as they note, we admit this in the discussion. We will add more to this text and stress that this paper simply adds to the evidence that daytime measurements of reef metabolism may not be enough to discern changes in reef health, especially on reefs with high algal cover.**

**We endeavoured to provide some nighttime measurements and admit that the replication is not enough. We will be clear in the discussion that nighttime measurements are important so flow metabolism teams should be constructed so that half the team sleeps during the day. Admittedly, we wanted to take more nighttime measurements but were flat out doing work on the reef all day for multiple different studies.**

Specific comments

Abstract

L 23- erase coma after other.

**Noted, will fix.**

Introduction

L83-86- The authors should be more specific here by including the changes in coral cover after the bleaching event (from 7.1% to 5.8% coral cover).

**Thank you for these details, they will be added to the text.**

L88-94- Authors should mention that in the same paper, Kayanne also observed decreases in NEC following a bleaching event and decreases in coral cover in Palau.

**We agree with this comment and will be adding the Palau results with the discussion of other publications showing a bleaching-driven decline in NEC (L56-65)**

Materials and Methods

L 159-170- Authors need to provide more details about NEP and NEC calculations. This section is oversimplified specially compared to section 2.1. This information needs to be in the main manuscript, not in the SI.

**Noted, much of the SI will be moved to the main text methods.**

Why are nighttime measurements not included in this section?

**Nighttime measurements will be added to this section.**

Discussion

L253-255- The authors previously mentioned that the NEP values were not included for the slack-water approach given the large source of error in air-sea oxygen exchange. Therefore, they should not consider this data in the discussion.

**The NEP values listed here are from the Eulerian approach which are collected close in time and do not exhibit the same error.**

L 259 -Again, authors should mention the contrasting results reported by Kayanne et al 2005 between the Palau and Japan studies.

**Noted, this will be added.**

L 322-329 -This information is not accurate. Courtney et al did not **find** that the dissolution signal was a major driver of the 24-h zero NEC signal during bleaching. They **Hypothesize** that reductions in NEC could influence carbonate dissolution and they link this hypothesis to the zero NEP observed during bleaching. Please, make the appropriate changes.

**We will change find to hypothesize. Thank you.**

Overall, I agree with the main point of the paper that NEC measurements may no longer be a good proxy for reef growth in degraded coral reefs. However, this is especially true for daytime measurements. During daytime, other benthic groups such as algal turfs, which are becoming more frequent as reefs degrade, can highly influence the metabolic signal of the reef. For example, Romanó de Orte et al. (2021) recently showed that, during daytime, algal turf communities have similar calcification rates than live corals. However, during nighttime,

while corals are still net calcifying, algal turfs are net dissolving. This would likely influence NEC during a 24 hours cycle. It is crucial to have robust nighttime NEC measurements in order to access changes in NEC during bleaching events. Further, the algal turf calcification during daytime could also help explain the discrepancies described in L343-364.

**We are aware of De Orte et al. (2021) and as noted with the other reviewers, we will be adding a similar back of the envelope calculation used in Figure 4 (modeled thermally-accelerated calcification) to project how the dominant fleshy algae cover on dead coral could have masked daytime declines in coral NEC. We agree it would be ideal to have more robust measurements of full 24-h NEC.**

L 360 Erase "be"before influence

**Noted, will fix**

**Note to Editor:**

**The driving need for these types of studies, in short, is to help the world find a way to measure coral reef health in the face of climate change.**

**We would like to add text to clarify that the specific emphasis behind this study is the development of marine ecosystem condition indicators for ocean accounting (https://www.oceanaccounts.org/). This is an international initiative by UN-member nations to advance environmental accounting procedures into the marine environment in response to the recently adopted SEEA (System of Environmental Economic Accounting). There is a specific focus in this initiative to find approaches that define marine ecosystem health and can be repeated by scientists and citizens trained by scientists around the world to be incorporated into environmental accounts maintained by governments, especially in underdeveloped nations where many coral reefs are located.**

**We will make every effort to admit more robust approaches (nighttime, dawn, dusk) will be needed to fully constrain 24-h NEC and we appreciate the reviewer's insight into these needs. We completely agree, from a scientific standpoint, that pairing nighttime measurements is extremely important. But we would also like to discuss that such requirements could prevent the application of this method on reefs which are hard to access at night due to navigation, technology, or safety constraints.**

**Altogether, we feel it is important to keep the bigger picture in mind of "how do we measure reef health throughout the world?" and would like to discuss that if daytime NEC cannot adequately do this on degraded reefs and nighttime measurements are required, this may limit the coral reef accretion estimates by flow metabolism in less developed nations who do not have access to autonomous water samplers or on coral reefs which are inaccessible at night.**

---

## Author Response (AR1)

**Will daytime community calcification reflect reef accretion on future, degraded coral reefs?**

Coulson A. Lantz[1,2], William Leggat[2], Jessica L. Bergman[1], Alexander Fordyce[2], Charlotte Page[1], Thomas Mesaglio[1], Tracy D. Ainsworth[1]

[1]University of New South Wales, School of Biological, Earth and Environmental Sciences, Kensington, 2033 NSW Australia

[2]University of Newcastle, School of Environmental and Life Sciences, Callaghan 2309 NSW Australia

Email Corresponding Author: C.lantz@unsw.edu.au

**Reviewer Response Form**

**Reviewer Comment #1**

*This study presents an interesting idea that temperature-induced increases in metabolism of various non-coral calcifying organisms could offset the expected effects on coral calcification during bleaching events, and thus that ecosystem-based measurements of calcification may not fully represent disturbance to these degraded habitats. The data are interesting and are presented in this new and insightful light, but the manuscript needs clarification in many aspects, and especially further consideration of statistical power.*

*Methods: not enough detail is provided to understand the methods. The supplement helps, but is far too long in my view (the supplement itself is separated into the main sections normally in a paper (Methods, Results, Discussion), so it almost reads like a separate paper. Importantly, essential details to understand the study are in the supplement. For example, the description of the Eulerian approach to NEC and NEP is too brief in the main text. The reader needs to access the supplement to begin to understand what was done here. Additionally, even the supplement is lacking some details, like precisely which samples were used for upstream and downstream TA and DO, why there is a 3600/100 in the equations, how was residence time calculated (e.g., where was the current meter?), how was TA titrated, etc.*

**We thank Reviewer #1 for their comments. We understand the reviewer would like more of the methods in the main text and we have addressed this by moving all text from the supplementary material into the main text. We apologize for this issue, it was a carryover from a prior submission to the Journal of Applied Ecology (who declined to review) where main text had to be minimized. We recognize the value for BG readers to have all of these methods in the main text.**

**Specifically, we have now provided details of the placement of the current meter (L220) and added the location to Figure 1. We have now described the flow metabolism equations in detail (L240 – L306) and highlighted the core reference where these approaches were based on (Langdon et al., 2010).**

We have now also clarified on L477 that thermally accelerated calcification is not the main conclusion of this paper but rather a proposed explanation, among 2 others (algal calcification, nighttime dissolution). Since the submission of this paper a publication by De Orte et al., 2021 (Unexpected role of communities colonizing dead coral substrate in the calcification of coral reefs, 2021, L&O) has provided compelling evidence for our other proposed mechanism: inorganic precipitation in dead coral substrate caused by localised algal photosynthesis. Given the dominant cover of fleshy and turf algae growing on dead coral in this study, we have endeavoured to include more discussion on L540 as an another proposed hypothesis for the lack of any changes in daytime calcification. Finally, the third hypothesis of nighttime dissolution driving the decline in net community NEC, has been further elaborated on in L554

*Results/Discussion: given the relatively high variance and large error bars on the NEC rates, is it surprising that there were not significant differences observed before/after bleaching, especially given the low number of independent samples (days)? An assessment of statistical power would be highly useful. Of course, it shouldn't be concluded that bleaching doesn't affect NEC. Rather, this study did not reject the null hypothesis that bleaching has no effect on NEC. It's a key distinction, one that is glossed over and somewhat misinterpreted here.*

Perhaps these concerns will benefit from having moved the methods from the SI to the main text. There were triplicate transects for 2 different reef zones measured daily. Furthermore, two separate flow metabolism approaches were used to estimate NEC, which is rare for coral reef community metabolism studies.

*Throughout: more clarity needed in how quantities were calculated and exactly how each type of data was used. For example, the text section 3.2 describes satellite SST begin to accumulate heat stress in February and refers to Figure 1. But from what I understand of the caption of Figure 1, only the in situ logger data are shown in that figure.*

This confusion is understood. Figure 1 displays the in-situ logger data (black line) and the accumulation of heat stress in these data (red line). The text discusses that accumulation of heat stress in the satellite data was the overall driver to begin this study. We have added a clarification to the legend of Figure 1 that the figure is based on data from temperature loggers. In the results, we have added text to L336-341 that describes the lagoon logger temperature data and differentiates these data from satellite data discussed on L372.

*Need to describe statistical approach in main text.*

Statistical methods have now been moved from the SI to the main text on L316.

*Throughout: it seems odd to say "community NEC" — usually, it's either "net community calcification" or "net ecosystem calcification"*

We understand this point. The word "community" was used to help differentiate from organism-level calcification. Per the reviewers suggestion, we have removed the word "community" before any instance of NEC to reduce redundancy. This is now defined early on (L14) that NEC and NEP speaks to the overall ecosystem. Organism NEC has now been changed to Net Organism Calcification (NOC) and this is defined on L488 and the equation has been changed to reflect this on L490.

*Throughout: need to decide if there is a space between numbers and % symbols or not.*

**Thank you for catching this. A space has now been added between numbers and % symbols throughout the manuscript.**

*Introduction: the discussion of existing literature is good and thorough, but perhaps there should be clearer differentiation between the effects of ongoing bleaching vs bleaching-induced mortality. Currently, the text describes these similarly, but it seems likely there would be different NEC responses to bleached (but living) corals as opposed to dead corals.*

**We understand this concern and it is due to, in part, the lack of literature across all of these scenarios. For this reason, the introduction attempts to draw from all of these examples to explain the overall progress, to date, in catching bleaching events in real time with NEC estimates. We would like to point out, however, that we make the effort to be specific about bleached (but living) corals vs. dead corals in the thermal-acceleration calculations for Figure 4. We direct the reviewer to L473 where we use existing literature to define what the potential calcification rate of a bleached, but not dead coral could be (~ 60 % reduction; D'Olivo & McCulloch, 2017).**

*Line 88: Didn't Kayanne also observe a decline in NEC after bleaching in Palau though?*

**This is correct, Kayanne et al., 2015 found differing responses at different reefs. We endeavoured to recognize this reviewer's work and others earlier in the introduction where we established that the expected response is a decline in NEC in response to bleaching (L54 – 65). We have added the Kayanne Palau results to this evidence as well on L65. Lastly we have edited the text on L94 to more clearly define that lack of a bleaching effect in Kayanne et al., 2015 was specifically on Shiraho reef.**

*Lines 88-89: again, bleaching vs bleaching-induced mortality seem to be conflated. Kayanne describe changes after bleaching-induced mortality, which the present text is comparing to NEC during a bleaching event (but with still-living corals). More clarity is needed about the difference between the two.*

**We understand the reviewer's point of conflating NEC during bleaching vs. after bleaching, but studies are so limited of either situation that we instead use the introduction to simply introduce any and all studies with NEC and bleaching through the lens of described coral cover. This critical difference between during bleaching vs. after bleaching is discussed on L564 and L523 (60% reduction in calcification for live, bleached corals) in the context of the results here.**

*Line 134: should state how many points were used on each image, and if the points were randomized.*

**There are no points on the image. The point-contact method records the cover and the photo-quadrat method uploads the image to ImageJ, the scale is set using the 1 m side of a photo quadrat, and the tracing tool is used to quantify the relative area of the benthic categories and differentiates between bleached or unbleached coral and clean or overgrown sediment. Text has been added on L150-153.**

*Line 152: "using using"*

**Fixed, thank you.**

*Line 232: delete "extremely"*

**Fixed, thank you**

*Figure 4: why not just have 2 bars for each temperature?*

**The figure is shown as one bar to illustrate the relative change in each benthic category as temperatures increased. We feel 2 bars would add more clutter. The change in the polygon of each category helps illustrate the relative increase or decrease.**

**Thank you for your comments!**

**Reviewer Comment #2**

**General Comments**

Overall, the theory of the study is great, and inter-disciplinary work like this is great to see. However, the linking of study elements (for example, quantification of the relationship between Symbiodiniaceae densities or photosynthetic yield and NEC) could be more deeply explored. Importantly, I find the methodology lacking necessary information to determine the validity of the results and many facets of the methodology and further analyses require justification. In the supplementary material, the equations used to calculate metabolic rates are not well defined and in their current state may be incorrect. The authors should take care with the accuracy of information presented from the literature and with the appropriateness of citations to fit the manuscript's narrative. I find the results interesting, but their main point seems oversold and broadly declarative with the data that is presented.

**We thank Reviewer #2 for their comments. We have moved all text in the supplemental material to the main document to address concerns regarding necessary information to repeat the study. We have corrected citations where necessary but also believe that the equations are correct in their current state and follow the approach outlined by Langdon et al., 2010. We would like to clarify that the thermally-accelerated calcification is not the main conclusion of the paper, but 1 of 3 proposed hypothesis to explain the main conclusion of the paper, which is that daytime NEC did not respond to a bleaching event. We have now elaborated more on the other 2 hypotheses (turf algal calcification [De Orte et al., 2021], and nighttime dissolution) on L540.**

**Specific Comments**

*Line 67-69 this information is incorrect. The bleaching event year is 2016, and the 2016 survey by Pisapia et al. cited here occurred after the bleaching event occurred in 2016. The 2018 survey showed an increase in NEC compared to 2016, but was still depressed related to pre-disturbance rates (see Abstract and Table 1 in cited paper).*

**Thank you for catching this. This is a typo, the text should read 2016. This text is straddling two studies (Pisapia et al., 2019 and McMahon et al., 2019). We have changed the text to 2016.**

*I disagree with the sentence from Line 74-76: A community transitioning to algal dominated from a coral dominated community would likely demonstrate changes in NEC and NEP, as indicated by many prior studies. The citation here (Courtney et al., 2018) is inappropriate because this paper does not indicate that NEC will become less effective with reef state transitioning. Rather, the authors state ". . . bleached coral reefs that recover quickly likely experience ephemeral reductions in reef NEC while systems shifting to alternative non coral-dominated states are likely to face lasting decreases in NEC." Reduced or low NEC, regardless of the cause or the dominant benthic class, is what is useful when investigating reef state.*

**The purpose of this text is to explain that if anthropogenic stressors continue to increase the ratio of algae:coral on reefs, then NEC may no longer be representing reef accretion by hermatypic coral but rather the summation of daytime NEC by organisms not actually creating reef structure. This idea is proposed by Courtney et al. 2018 in the discussion, so we believe the citation is appropriate:**

*"In contrast, measurements of NEC at Shiraho Reef, Japan did not change during the September 1998 bleaching event, where 51% of the total 7.1% total coral cover was bleached compared to a recovery survey conducted in September 1999 with 6.7% total coral cover and no bleaching observed (Kayanne et al. 2005). Kayanne et al. (2005) hypothesized that calcification by living bleached corals, calcifying algae, and benthic foraminifera may have compensated for bleaching-induced losses in NEC at Shiraho Reef. Indeed, the dominant calcifiers of coral reefs include corals, red coralline algae, molluscs, green calcifying algae, and benthic foraminifera (Montaggioni and Braithwaite 2009), but their relative contributions to coral reef CaCO3 budgets and how these change under different reef states are uncertain. This raises the question and need to further quantify the relative importance of contributions by other calcifiers to coral reef NEC especially for low coral cover (< 10%) and bleached coral reefs."*

*Line 84-85, this statement is incorrect. The Kayanne et al. 2005 abstract states 'All the metabolic parameters, Pg, R, E and calcification (G) were reduced by half after the bleaching,', and no pre-bleaching rates were estimated.*

**In addition to the text above from Courtney et al., 2018 summarizing Kayanne et al., 2005 Shiraho Reef results, there is also specific text in the Kayanne paper which explains differing results between Palau and Japan. Yes, the abstract states G was reduced by half but that is for Palau. Within the text they describe differing responses in Palau and Shiraho reef at Ishigaki Island, Japan. The following is taken specifically from Kayanne et al., 2005 section 3.1.4:**

*However, E was lower in September 1998 (36 mmol m_2 d_1); this was during the bleaching period, when the mean coverage of living coral was 7.1% (51% of which was bleached, so that the coverage of corals with symbiotic algae was 3.6%). After the bleaching, E increased over the course of recovery. By contrast, G (calcification) remained nearly constant during the September 1998 bleaching at Ishigaki Island.*

**We fully admit starting on L66 that the normal expectation is for bleaching-induced decline in NEC. We have added Kayanne's Palau results to further support this expectation. When we found no change in NEC, we performed an exhaustive literature review to find any possible examples where this was also observed.**

Methods:

*Overall, I found the methods do not provide enough information or justification to assess the validity of the rates or to undertake a follow-up survey. One major concern is that the time window stated (1100-1500) is not representative of calcification over a diel cycle (which is what is typically used when discussing reef state). Different stressors can affect calcification/dissolution differently depending on the time of day. Prior studies have shown that impacted corals show a significantly higher rate of low-light dissolution than non-impacted corals, even when daytime or peak sunlight calcification is not affected.*

**Methods text has now been moved from the SM to the main text so that follow-up surveys may be conducted.**

**We agree that calcification can differ depending on time of day and light. Measurements on the Heron reef flat can only be conducted during a 3 hour period around low tide. This**

is common for work on tidal reef flats. Outside this period, there is too much water on the reef and mixing with the offshore water to measure adequate changes in water chemistry. Therefore, the tides must be followed and one cannot make multiple measurements on the same day.

It is important to note that the purpose of this study was not to describe the diel calcification trends on Heron reef flat, this has already been done multiple times, most recently by Stoltenberg et al., 2020. The purpose was to compare pre-bleaching and during-bleaching NEC using two different flow-metabolism approaches. Midday at full light provides the best opportunity measure peak community metabolism and ensure changes in seawater chemistry were large to reduce error. We admit in the discussion as a follow up more measurements need be conducted now at dawn, dusk, and night.

Why were coral fragments gathered for PAM fluorometry rather than assessment in situ? More information is needed on PAM measurements. How were those fragments chosen, how many of 'bleached' and healthy? Was there a control for PAM measurements or Symbiodiniceae densities during the bleaching? Were pre-bleaching yield or Symbiodiniceae densities measured, it does not appear so in Fig. 3.

More information has been provided on exactly how coral fragments were gathered on L185-190. In short, the listed taxa were gathered across the reef during the bleaching event. We gathered live branches of all these taxa to see if they were healthy or not from the standpoint of photophysiology.

The model PAM listed is a benchtop unit, we did not have an underwater PAM, so corals could not be measured in-situ.

It is important to note that examples of a flow metabolism study setting up all of these type of control measurements before the bleaching begins are extremely rare given our predictive abilities are generally not dependable enough to mobilize an entire flow-metabolism team and gear. Most studies (e.g., McMahon et al., 2019, Courtney et al., 2018, Pisapia et al., 2019) measure either during or after a bleaching event.

This study did not begin with a certain expectation of bleaching, it was initially started as a study to relate community-level NEC and NEP to the census approach from benthic surveys. In the middle of this satellite data indicated accumulation of heat stress and signs of bleaching began. Realizing this opportunity, the study objectives were quickly shifted to take advantage of this event. We recognized that qualitative examples of bleaching (photo quadrats, white corals) may not be enough to prove the coral's health was compromised and decided some physiology measurements (PAM, symb. Densities) would be a great way to add strength to this statement. Text has now been added to L180-182 to explain why this "opportunity" arose and why pre-bleaching physiology measurements were not available.

*Equation 1 & 2 in the Flow Respirometry Approach: I am making assumptions here because I don't understand these equations. If u u is current speed (cm s$^{-1}$)(not stated), then in the context of these equations, you are multiplying current speed by 3600 to transform from per second to per hour and dividing by 100 to transform from cm to m. However, if I am correct, this means that you have divided out your length component (m/m). The residence time is the term that needs to be on the denominator to provide the unit of mmol/m$^2$/hr. If I am incorrect here, then*

*more definition needs to be put into your equations to show that you are calculating the correct values. Typically, residence times are calculated separately than metabolic values using transect length, current speed, and length/time averaged depth (See Supplementary Information in DeCarlo et al., 2017, or Davis et al., 2020).*

**We have moved all of the SM methods text on these equations to the main text. See L265 for definition of current speed. For all other questions regarding the equations, we highly encourage the reviewer to read Langdon et al., 2010 (Langdon, C., Gattuso, J.-P., Andersson, A., Océanologique, O. and Pierre, U.: Part 3 : Measurements of CO 2 - sensitive processes 13 Measurements of calcification and dissolution of benthic organisms and communities Part 3 : Measurements of CO 2 - sensitive processes, , 213–232, 2010.)**

**This is an excellent book chapter which details how to conduct various community metabolism approaches, including Slack-water, Eulerian, and Lagrangian and is one of the few to detail the calculations needed (such as 3600/100 ) to correctly cancel out units in the equation to achieve a rate which details unit area per time.**

'Slack water' sampling is not made under the correct conditions. Slack water indicates no (or very slow) water movement, so this may be an issue of semantics and you really mean Eulerian sampling (see Silverman et al., 2014 and McMahon et al., 2019 for examples and comparisons). More information is needed here. If you are looking at changing water chemistry in the same place, you need either 1) and end member (an initial value), or 2) More specific calculation of your depth-averaged residence time over space.  Did you have a current meter? If so, where was it placed? Maybe add to site map?

**Overall, the slack-water name sounds like water isnt moving, but there are multiple studies (notably Langdon et al., 2010) which define that slack water simply means the water is contained in a basin and circulated within. This method has been used previously in the exact same location by Stoltenberg et al., 2020 (Late afternoon seasonal transition to dissolution in a coral reef: An early warning of a net dissolving ecosystem? GRL) and at nearby One-Tree island by Shaw et al., 2012 (Impacts of ocean acidification in naturally variable coral reef flat ecosystems, JGR). It has been traditionally employed on reef flats that are separated from the open ocean at low tide even if there is still water moving within the reef flat.**

**You do not need a current meter for the Slack-water approach, that is for Eulerian (which we also did). We wanted to compare two different approaches and the slack water with no current meter is the most commonly used on the Heron reef flat by prior studies.**

**Concerning the Eulerian approach, where we do need depth measurements and current speed, we direct the reviewer to L240 and Table 2, where the methods and results of these measurements are detailed.**

*'Slack water' Methods state: water samples were collected from the same three locations (n = 3 day-1) two hours before 84 peak low tide and one hour following." Where were these locations and at what interval were they taken? Were they taken at the same time/place as the 'flow' samples were taken?*

**The slack-water interval is 3 hours (two hours before, one following low tide to ensure separation from open ocean = 3 hour interval). The eulerian interval is as close in time as**

possible, as fast as we could walk 200m across the reef (upstream and downstream sample < 5 min apart). The locations of these sampling locations is noted in Fig. 1. This is detailed on L286 in a section with approach comparisons, but in general the downstream samples for the eulerian approach is the same location of the slack-water samples (which are collected in the same location 3 hours apart).

We thought that the effort of using both the Eulerian and Slack-water approach would help the presentation of the results, as it is rare to compare both approaches simultaneously to provide strength to community metabolism estimates. To prevent confusion with other readers, we have been more explicit with this in the main text to outline the Eulerian approach (L240), the Slack Water approach (L269), and text explaining the upside and drawbacks of each (L286)

*Figure 4: More information is needed on where the calcification rates were taken from in the literature. How were differences in calcification rates determined for different specific temperatures? Is this what's described in L285 – 288? If so, is the 1.1 degree change in temperature determined from absolute differences in temperature or for corals which bleached after 1.1 degree increase? If the bleaching temperature was 29.1, the calcification rate indicated here at 29.1 degrees should represent calcification rates under bleaching. Please specify.*

We have now moved these citations and how these rates were taken from the Supplemental Material to the main text on L494

Per the description of Figure 4, the calcification rate for coral indicates the rate expected based on coral NEC at 29.1 x reduced calcification rate expected under bleaching x the amount of coral bleached on the reef. This stated as follows, for example, on L472: "Further, if 60 % of the total coral cover was calcifying roughly 60 % slower due to bleaching (D'Olivo & McCulloch, 2017), this would imply that active calcifying coral cover was likely reduced to only 2 – 4 %." In other words, 60 % of the coral on the reef flat bleached. And the D'olivo paper suggests bleached coral calcifies 60 % slower. They are coincidentally the same number which may add to the confusion.

*Lines 306–308: How does this 9.8% expected decline in NEC compare with your observed results?*

There was no significant changes in NEC (Table 4), so its a 9.8% larger decline than observed. This is stated on L524

*L 312–316: This is a good description of this result, and it is an interesting result. However, I think the wording in the abstract and conclusions are too strong with the data provided to support the argument.*

We are not sure which result/conclusion exactly the reviewer refers to here. If its the conclusion regarding daytime NEC not actually reflecting coral accretion rates, we would contend there is enough information given we used 2 different flow metabolism approaches over 20 days across triplicate transects and these findings match well with recent work by De Orte et al., 2021, which showed dead coral calcifies just as fast as live coral during the day. There is a growing consensus that daytime NEC may not be

reflecting what we long have thought it does (reef accretion), especially on degraded reefs with very little coral cover.

If they are referring to the thermally-accelerated calcification conclusion, we would like to clarify that, overall, the unobserved decline in NEC despite observed bleaching provided the opportunity to hypothesize 3 different drivers behind why this occurred: 1) Thermally-accelerated calcification 2) Algal/Dead Coral calcification 3) Increased nighttime dissolution (or a combination of all 3). On L477 we discuss the thermally accelerated calcification in unbleached sessile calcifiers. Since the submission of this manuscript, a publication by De Orte et al., 2021 (Unexpected role of turf algae communities colonizing dead coral substrate in the calcification of coral reefs, 2021, L&O) has provided compelling evidence for our other hypothesis outlined in L540 (dead corals with algae might be calcifying) as well as the importance of nighttime dissolution measurements.

Technical Corrections

*Fig.7 is referenced a few times but no figure 7 is included, change to Fig. 4.*

We found 2 instances of a Fig. 7 reference and have changed these to Fig. 4

*Information in the table 2 caption should be placed in the methods.*

The information in Table 2 has now been moved to the methods. We have also added more information on how depth and flow were collected and compared. Altogether, text regarding depth has been added to L220-222. Text explaining the placement of the ADV current meter has been added to L219 and text explaining how the use of flourescein dye in combination with the ADV was used to establish an understanding of an acceptable flow regime and time period for the Eulerian and Slack Water approach on L246-260

Thank you for your comments!

General comments:

This is a timey study that discusses the possible divergence between estimates of NEC and reef growth in degraded coral reefs. The authors provide an interesting perspective that thermal enhancement of calcification in other benthic members may highly influence NEC, especially in reefs where coral cover is low.

The limited amount of nighttime NEC measurement is a weakness of the study. Nighttime dissolution could significantly influence the 24 hours NEC signal, especially if other benthic groups are contributing substantially to the calcification signal. The authors do a good job discussing this issue in the "future considerations" section. However, the lack of these data could have influenced the conclusions of this study.

There are important information missing in the main text while the SI is too long. Authors could change the structure of the paper by including some sections of the SI (e.g. S2.2.) in the main manuscript.

Overall the paper is well written but there are some references and details missing which are highlighted in the specific comments.

**We thank reviewer #3 for their review of this manuscript. Their concerns fall in line with Reviewer #1 and #2, requiring more text from the SI to moved to the main text which has been rectified.**

**We also agree that nighttime dissolution could be a major driver of the NEC decline that we did not measure and, as they note, we admit this in the discussion. We have added more to this text and stress that this paper simply adds to the evidence that daytime measurements of reef metabolism may not be enough to discern changes in reef health, especially on reefs with high algal cover (L538 and L615).**

**We endeavoured to provide some nighttime measurements and admit that the replication is not enough. We will be clear in the discussion that nighttime measurements are important so flow metabolism teams should be constructed so that half the team sleeps during the day or that these data are collected on reefs accessible at night. Admittedly, we wanted to take more nighttime measurements but were flat out doing work on the reef all day for multiple different studies.**

Specific comments

Abstract

L 23- erase coma after other.

**Fixed**

Introduction

L83-86- The authors should be more specific here by including the changes in coral cover after the bleaching event (from 7.1% to 5.8% coral cover).

**The decline to 5.8% has been added to L96**

L88-94- Authors should mention that in the same paper, Kayanne also observed decreases in NEC following a bleaching event and decreases in coral cover in Palau.

**This observation has been now added to L66**

Materials and Methods

L 159-170- Authors need to provide more details about NEP and NEC calculations. This section is oversimplified specially compared to section 2.1. This information needs to be in the main manuscript, not in the SI.

**A large portion of the SI has now been moved to the main text.**

Why are nighttime measurements not included in this section?

**Nighttime measurements have now been added to this section.**

Discussion

L253-255- The authors previously mentioned that the NEP values were not included for the slack-water approach given the large source of error in air-sea oxygen exchange. Therefore, they should not consider this data in the discussion.

**The NEP values listed here are from the Eulerian approach which are collected close in time and do not exhibit the same error.**

L 259 -Again, authors should mention the contrasting results reported by Kayanne et al 2005 between the Palau and Japan studies.

**The structure of the paper is to first discuss examples where the expectation occured (decline in NEC and coral cover). Kayanne et al 2005 has been added as a reference for this part (L66). The next paragraph then delves into the nuances of these findings and provides examples where the expectation was not met and proposes the underlying potential drivers (low coral cover; L94).**

L 322-329 -This information is not accurate. Courtney et al did not **find** that the dissolution signal was a major driver of the 24-h zero NEC signal during bleaching. They **Hypothesize** that reductions in NEC could influence carbonate dissolution and they link this hypothesis to the zero NEP observed during bleaching. Please, make the appropriate changes.

**We have changed find to hypothesize (L539)**

Overall, I agree with the main point of the paper that NEC measurements may no longer be a good proxy for reef growth in degraded coral reefs. However, this is especially true for daytime measurements. During daytime, other benthic groups such as algal turfs, which are becoming

more frequent as reefs degrade, can highly influence the metabolic signal of the reef. For example, Romanó de Orte et al. (2021) recently showed that, during daytime, algal turf communities have similar calcification rates than live corals. However, during nighttime, while corals are still net calcifying, algal turfs are net dissolving. This would likely influence NEC during a 24 hours cycle. It is crucial to have robust nighttime NEC measurements in order to access changes in NEC during bleaching events. Further, the algal turf calcification during daytime could also help explain the discrepancies described in L343-364.

**We agree, De Orte et al. (2021) is an excellent paper. This was published after the initial submission of this manuscript, and we now discuss it would be ideal to have more robust measurements of full 24-h NEC (L617). This is an important takeaway for managers and policy makers alike who can use the findings to support why they may need to use different approaches (census-based or satellite-based) to monitor bleaching events in reef locations not accessible at night.**

L 360 Erase "be" before influence

**Fixed**

**Thank you for your comments!**

---

## Referee Report (RR1)

The authors did a good job revising their manuscript and have satisfied most of the reviewer comments. I have a few minor suggestions below.

The authors seem to suggest that only coral calcification contributes to reef formation. My understanding is that other calcifiers, coralline algae in particular, also produce $CaCO_3$ that contributes to reef structure. It may be helpful to talk more about losing complex 3-D habitat if corals are replaced by other calcifiers, but I will leave this to the authors' discretion.

Line 29: "corals… are a… species": Plural vs singular mismatch?

Line 31: not sure "secrete" is the best verb: "precipitate", "build", "construct", "form", etc. would be better

Throughout: typically, the word "which" only comes after a comma. There are many instances where the authors use "which" but that in my view it should be "that".

Line 57: delete "that"

Line 214: punctuation typo here

Proper spelling is "Hawai'i", not "Hawai'i"

---

## Author Response (AR2)

**Will daytime community calcification reflect reef accretion on future, degraded coral reefs?**

Coulson A. Lantz[1,2], William Leggat[2], Jessica L. Bergman[1], Alexander Fordyce[2], Charlotte Page[1], Thomas Mesaglio[1], Tracy D. Ainsworth[1]

[1]University of New South Wales, School of Biological, Earth and Environmental Sciences, Kensington, 2033 NSW Australia

[2]University of Newcastle, School of Environmental and Life Sciences, Callaghan 2309 NSW Australia

Email Corresponding Author: C.lantz@unsw.edu.au

**Reviewer Response Form**

**Reviewer Comment #1**

*The authors did a good job revising their manuscript and have satisfied most of the reviewer comments. I have a few minor suggestions below.*

**We thank Reviewer #1 for their follow up review of the manuscript and feel that the manuscript benefits greatly from their dedicated comments and editing.**

*The authors seem to suggest that only coral calcification contributes to reef formation. My understanding is that other calcifiers, coralline algae in particular, also produce CaCO3 that contributes to reef structure. It may be helpful to talk more about losing complex 3-D habitat if corals are replaced by other calcifiers, but I will leave this to the authors' discretion.*

**This is a good point and it is important to clarify we mean the loss of 3D structure and complex habitat.**

**In the abstract on L24 we have added "complex, three-dimensional reef structure" to clarify the type of accretion loss.**

**On L492, we have added the sentence "Although some of these calcifiers still accrete limestone structure (e.g., coralline algae), none replace the complex, three-dimensional structure uniquely created by corals."**

**On L555, we have clarified the reduced capacity for "complex, three-dimensional structure"**

*Line 29: "corals… are a… species": Plural vs singular mismatch?*

**Thanks for catching, we have changed to "coral....are a....species"**

*Line 31: not sure "secrete" is the best verb: "precipitate", "build", "construct", "form", etc. would be better*

**Agreed, we have changed to "construct"**

*Throughout: typically, the word "which" only comes after a comma. There are many instances where the authors use "which" but that in my view it should be "that".*

**All instances where "which" did not come after a comma were changed to "that".**

*Line 57: delete "that"*

**Deleted**

*Line 214: punctuation typo here*

**Edited into 2 separate sentences.**

*Proper spelling is "Hawaiʻi", not "Hawaiʼi"*

**Thanks for identifying the need for an Okina, not an apostrophe. We have inserted " ` " instead of " ' " to represent the Okina. Perhaps this can be addressed in the preprint process.**